# PYRAMID VECTOR QUANTIZATION FOR LLMS

## ABSTRACT

Recent works on compression of large language models (LLM) using quantization considered reparameterizing the architecture such that weights are distributed on the sphere. This demonstratively improves the ability to quantize by increasing the mathematical notion of coherence, resulting in fewer weight outliers without affecting the network output. In this work, we aim to further exploit this spherical geometry of the weights when performing quantization by considering *Pyramid Vector Quantization* (PVQ) for large language models. Arranging points evenly on the sphere is notoriously difficult, especially in high dimensions, and in case approximate solutions exists, representing points explicitly in a codebook is typically not feasible due to its additional memory cost. Instead, PVQ uses a fixed integer lattice on the sphere by projecting points onto the 1-sphere, which allows for efficient encoding and decoding without requiring an explicit codebook in memory. To obtain a practical algorithm, we propose to combine PVQ with scale quantization for which we derive theoretically optimal quantizations, under empirically verified assumptions. Further, we extend pyramid vector quantization to use Hessian information to minimize quantization error under expected feature activations, instead of only relying on weight magnitudes. Experimentally, we achieve state-of-the-art quantization performance with pareto-optimal trade-off between performance and bits per weight and bits per activation, compared to competitive methods. On weight-only, we find that we can quantize a Llama-3 70B model to 3.25 bits per weight and retain 98% accuracy on downstream tasks.

## 1 INTRODUCTION

Quantization enables compression of large language models (LLMs) by reducing the number of bits per weight required to represent weights. Weight outliers can make quantization difficult, as they cause weight distributions to not match the implicit evenly distributed grids used in many quantization methods. To overcome this, recent works have proposed to reparameterize architectures by rotating weights in a way that leaves the network as a function unchanged (Ashkboos et al., 2024; Chee et al., 2024).

In this work, we aim to exploit the spherical geometry in weights when performing quantization by considering Pyramid Vector Quantization (PVQ) (Fischer, 1986) for LLMs. In PVQ, weights are quantized on a hyper-pyramidal lattice that allows efficient encoding and decoding without having to explicitly represent a codebook in memory. By projecting the lattice onto the hyper-sphere, a quantization grid is obtained that accurately approximates a uniform grid on the spherical domain.

PVQ has been very successful in well-known audio (Valin et al., 2012) and video codecs (Daede et al., 2016). We demonstrate that the same algorithm allows practical quantization of large language models, by proposing a group-wise quantization scheme and further extending PVQ to use Hessian information accounting for curvature in the loss. We also propose a scheme to quantize the normalized scales (amplitudes) of each group according to theoretically derived quantiles, which we verified to closely match the empirical weight distributions of pretrained LLMs in practice. Our contribution extends beyond Liguori (2017), which considered PVQ on weights of small neural networks.

Experimentally, we find that our proposed PVQ quantization scheme outperforms the state-of-the-art in terms of bits per weight and bits per activation. We do not only perform simulated quantization, but also provide kernels that allow hardware accelerated encoding and decoding of PVQ. We achieve state-of-the-art quantization on the most prominent Llama-3, Phi-3 and Mistral architectures in terms of performance against bits per weight (BPW). In particular, we demonstrate 3.25 bit weight quantization at a negligible 1-3% drop in performance, as measured in accuracy on downstream tasks.

## 2 BACKGROUND

Before discussing our approach on using pyramid vector quantization to quantize LLMs, we provide an overview of vector quantization, spherical geometry of LLM weights, and describe classic PVQ.

### 2.1 QUANTIZATION

Quantization is a compression technique for machine learning models by storing weights ($\boldsymbol{W}$) or activations ($\boldsymbol{x}$) in a some chosen lower bit representation, such as lower precision floats or scaled integers ($\widehat{\boldsymbol{W}}$). A common conversion is to minimize a second-order layer-wise proxy loss (Nagel et al., 2020; Frantar et al., 2022),

$$L(\widehat{\boldsymbol{W}}) = \mathbb{E}_{\boldsymbol{x}} \left[ ||\boldsymbol{W}\boldsymbol{x} - \widehat{\boldsymbol{W}}\boldsymbol{x}||_2 \right] = \mathrm{Tr}\left( (\boldsymbol{W} - \widehat{\boldsymbol{W}})\boldsymbol{H}(\boldsymbol{W} - \widehat{\boldsymbol{W}}) \right) \tag{1}$$

where the layer-wise Hessian $\boldsymbol{H} = \mathbb{E}_{\boldsymbol{x}} \left[ \boldsymbol{x}_n \boldsymbol{x}_n^T \right] \approx \frac{1}{N} \sum_{n=1}^{N} \boldsymbol{x}_n \boldsymbol{x}_n^T$ is empirically estimated using a calibration dataset $\{\boldsymbol{x}_n\}_{n=1}^{N}$. The objective is optimal in the sense that it minimizes the layer-wise output at each layer, but a crude approximation with respect to the actual training loss. Some recent works have proposed to also use gradients to improve the approximation to the true training objective (van der Ouderaa et al., 2023). Although it would be interesting to combine ideas presented in this paper with gradients, we stick to a layer-wise loss Equation (1) for simplicity and because this yields a faster quantization method that does not require backpropagation.

### 2.2 VECTOR QUANTIZATION

Instead of individually quantizing weights, as done in scalar quantization, vector quantization aims to simultaneously quantize multiple weights. It can be shown that, even for completely independent Gaussian sources, this typically results in much higher theoretical signal-to-noise ratios, leading recent works to consider vector quantization for LLMs (van Baalen et al., 2024; Liu et al., 2024b; Egiazarian et al., 2024; Tseng et al., 2024a;b). Yet, vector quantization is not widely adopted in practice because of two practical problems. Firstly, naive vector quantization requires constructing an explicit codebook using clustering (such as K-means), quickly becoming infeasibly large for higher number of dimensions. Secondly, quantization requires an expensive search over this explicit codebook, which can not practically be used on-the-fly on activations. Although application to LLMs is limited, it is common in vector quantization (Gray & Neuhoff, 1998) to use an **implicit codebook** which does not have to be explicitly instantiated in memory. PVQ is such a vector quantization and comes with the additional benefit of being **search-free**, allowing encoding and decoding of vectors without having to perform an explicit and exhaustive lookup that at least linearly scales in algorithmic complexity with the size of the codebook. As a result, it can be applied on-the-fly and not only on the neural network weights, but also to the activations during inference.

### 2.3 WEIGHTS ON THE SPHERE

Instead of viewing weight vectors in Euclidean $D$-space $\boldsymbol{w} \in \mathbb{R}^D$, they can be interpreted as scaled points $\boldsymbol{w} = s\boldsymbol{v}$ on the unit sphere $\boldsymbol{v} \in \Omega_D$, with $\Omega_D = \{\boldsymbol{v} = (v_1, v_2, \ldots, v_D) \in \mathbb{R}^D : ||\boldsymbol{v}||_2 = 1\}$. We refer to this spherical decomposition as the *direction* $\boldsymbol{v} = \frac{\boldsymbol{w}}{||\boldsymbol{w}||_2}$ and the *amplitude* $s = ||\boldsymbol{w}||_2$ of a vector. Recent works that use such a spherical perspective on LLM weights have offered new insights in properties of the training dynamics and guide algorithmic improvements. For example, (Kosson et al., 2023) noted that LLM weights under weight decay or popular deep learning optimizers converge to an equilibrium on the sphere, theoretically predicting the magnitude of the amplitude after training. Recent works have shown that only updating direction components can be beneficial

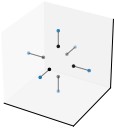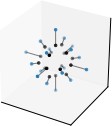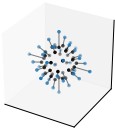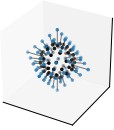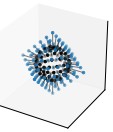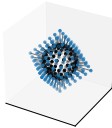

Figure 1: Illustration of the PVQ integer lattice in $d = 3$ dimensions with increasing pulses $k$ from 1 to 6. Points on the pyramid $\mathcal{P}_{3,k}$ are projected onto the sphere $\mathcal{S}_{3,k}$.

in low-rank adaptation Liu et al. (2024a) and training itself (Loshchilov et al., 2024). Yet, weights are not always uniformly distributed making quantization hard and giving rise to outliers. We use *coherence processing* (Chee et al., 2024) to reparameterize weights on the sphere in a way that reduces outliers without functionally changing the output of the network. Further, in high dimensions the information in the amplitude is negligible compared to the information in the direction (Kipnis & Reeves, 2021). The observation that LLM weights are in practice uniformly distributed across the sphere is the primary motivation behind exploring pyramid vector quantization, which allows us to construct an efficient spherical quantization code.

**Hadamard coherence processing**  Recent works have shown that rotating weights (Chee et al., 2024; Ashkboos et al., 2024) can improve weight coherence and reduce the number of outliers without altering the network's output, $\widetilde{\boldsymbol{W}} = \boldsymbol{U}\boldsymbol{W}\boldsymbol{V}$ where $\boldsymbol{U} \in \mathbb{R}^{R \times R}$ and $\boldsymbol{V} \in \mathbb{R}^{C \times C}$ are both orthogonal matrices. Since the transpose of orthogonal matrices equals their inverse, the reparameterisation can simply be undone in the forward pass by left- and right multiplying with matrix transposes, $\boldsymbol{W} = \boldsymbol{U}^T\widetilde{\boldsymbol{W}}\boldsymbol{V}^T$. We follow Chee et al. (2024); Ashkboos et al. (2024) and use random Hadamard matrices for $\boldsymbol{U}$ and $\boldsymbol{V}$, which can be implemented very efficiently.

Representing weights on the sphere through incoherence processing has shown to improve existing quantization methods by preventing outliers in weight distributions. Yet, these methods still quantize weights to Euclidean grids and do not exploit the spherical geometry of underlying weight distributions. We explore pyramid vector quantization to construct a quantization grid that is tailored to the spherical geometry by being approximately uniform on the sphere.

**Absorbing rotation matrices**  In most cases, the additional rotation matrices do not result in a memory or compute overhead during inference since rotation matrices can be absorbed into weight matrices: depending on the placement of the rotation matrix in the architecture, it can be left- or right- multiplied with an adjacent weight matrix. This principle was used in QuaRot (Ashkboos et al., 2024), which describes how rotation matrices can be efficiently absorbed into attention and fully-connected layers in commonly used LLM architectures.

## 2.4 CLASSIC PYRAMID VECTOR QUANTIZATION

To quantize points on the sphere, we would like to construct *spherical code*, a finite subset on the unit sphere $\mathcal{S} \subset \Omega_D$. Packing a set of points on the surface of a sphere such that their distance is maximized is a notoriously hard problem in mathematics, famously dating back to the Dutch botanist Tammes (1930). Just like sphere packing, optimal spherical codes are not generally known, with the exception of some specific dimensions, similar to the E8 packing in 8 dimensions exploited in (Tseng et al., 2024a). Even though good but sub-optimal spherical codes exist (Conway & Sloane, 2013), there is not always an efficient method to enumerate the packing without requiring an explicit codebook, resulting in costly quantization. PVQ (Fischer, 1986) provides a solution to both of these issues simultaneously, allowing good spherical codes to be constructed in arbitrary dimension that can be efficiently encoded and decoded without having to maintain an explicit codebook in memory. This is achieved by projecting an integer lattice on the $l_1$ ball $\mathcal{P}_{D,K}$ onto the hypersphere $\mathcal{S}_{D,K}$. We now formalize these concepts by providing an overview of classic PVQ algorithm.

**The integer pyramid lattice**  Formally, the integer lattice of PVQ on the $D$-dimensional hypersphere is obtained by starting from a set of points on the $l_1$ ball of radius $K$, and projecting the set

of points onto the sphere. We denote the set of integer points on the $l_1$ ball as $\mathcal{P}_{D,K}$,

$$\mathcal{P}_{D,K} = \left\{ \boldsymbol{p} \in \mathbb{Z}^D : ||\boldsymbol{p}||_1 = \sum_{d=1}^{D} |\boldsymbol{p}_d| = K \right\} \tag{2}$$

To obtain our spherical code $\mathcal{S}_{D,K}$, we can project the points $\mathcal{P}_{D,K}$ onto the sphere, by normalizing,

$$\mathcal{S}_{D,K} = \left\{ \frac{\boldsymbol{p}}{||\boldsymbol{p}||_2} : \boldsymbol{p} \in \mathcal{P}_{D,K} \right\} . \tag{3}$$

The number of codes $N(D, K) = |\mathcal{S}_{D,K}| = |\mathcal{P}_{D,K}|$ can be written as

$$N(D, K) = 2D \cdot {}_2F_1(1 - D, 1 - K, 2; 2) \tag{4}$$

where ${}_2F_1(a, b, c; z)$ is the hypergeometric function (Terriberry, 2007), and for specific $D, K \in \mathbb{Z}$ can be computed through the following recurrence relation,

$$N(D, K) = N(D - 1, K) + N(D, K - 1) + N(D, K) \tag{5}$$

where we define $N(D, 0) = 1$ for all $D \geq 0$ and $N(0, K) = 0$ for all $K \geq 1$.

## 2.5 SUBROUTINES OF CLASSIC PVQ

**Quantizing the direction** To quantizing a vector $\boldsymbol{w} \in \mathbb{R}^D$, we map it to the closest point on the pyramid $\mathcal{P}_{D,K}$ using an iterative procedure that projects the vector $\boldsymbol{w}_0 = \boldsymbol{w}$ onto the $l_1$ ball of radius $K$, and round it to the closest integer,

$$\boldsymbol{w}_{t+1} = \text{quantize\_step}_K(\boldsymbol{w}_t) = \text{round}\left( \frac{K}{||\boldsymbol{w}_t||_1} \boldsymbol{w}_t \right) \tag{6}$$

After this step, we check whether the norm satisfies $||\boldsymbol{w}||_1 = K$. If this is the case, we're done. If not, we either decrease by 1 or increase by 1, the element in $\boldsymbol{w}_t$ with the biggest deviation $|\boldsymbol{w}_i| - K$. We then call quantize\_step$_K(\cdot)$ again, and repeat this process until convergence, which should happen within at most $T < D$ steps. The resulting vector lies on the pyramid $\boldsymbol{w}_T \in \mathcal{P}_{D,K} \subset \mathbb{Z}^D$, meaning it is integer-valued and the absolute values sum to 1. We provide pseudocode of the quantization algorithm in Algorithm 2 in Appendix B.2 to be fully self-contained.

**Pyramid encoding** We can encode points on the pyramid $\boldsymbol{p} \in \mathcal{P}_{D,K}$ to integer codes $c \in [1, \ldots, N(D, K) - 1]$ using an efficient algorithm, which avoids having to build an explicit table in memory to represent the codebook. The original PVQ paper (Fischer, 1986) describes an encoding scheme, of which we provide pseudocode Algorithm 2 in Appendix B.2, for the purpose of being self-contained. The algorithm provides a bijective mapping from the points of the pyramid $\mathcal{P}_{D,K}$ to the set of integer indices $\mathcal{C}_{D,K}$, providing a compact representation that allow vectors to be efficiently stored in as few bits as possible. In Figure 2, we provide an example illustration of points on $\mathcal{P}_{2,7}$ and indices $\mathcal{C}_{2,7} = [0, 27]$. The table of $N(d, k)$ can be precomputed for $0 \leq d \leq D$ and $0 \leq k \leq K$ and reused to avoid recomputing the same quantity.

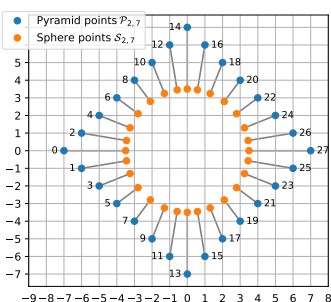

Figure 2: Illustration of points on pyramid $\mathcal{P}_{2,7}$, their projections onto the sphere $\mathcal{S}_{2,7}$ and codes in $\mathcal{C}_{2,7}$.

**Pyramid decoding** We decode the integer codes $c \in \mathcal{C}$ of PVQ to their associated vectors $\boldsymbol{p} \in \mathcal{P}_{D,K}$ through a decoding algorithm which performs the inverse operation of the encoding algorithm above. In Appendix B.3, we provide a corrected version of the decoding algorithm described in the original PVQ paper (Fischer, 1986). The original paper contains an error and misses a line after setting $x_i \leftarrow -j$, resulting in wrong decodings when $\boldsymbol{p}$ contains negative values, except for when the last value is negative (which happens to be the case for the example given in the original paper).

# 3 PVQ FOR LLM COMPRESSION

Before we present our overall algorithm to perform PVQ to LLMs in Section 3.3, we discuss the motivations and practical benefits of PVQ for LLM quantization in Section 3.1 and analyse the theoretical signal-to-noise ratio of PVQ Section 3.2.

## 3.1 PRACTICAL ADVANTAGES OF PVQ

PVQ offers several practical advantages over competing methods. Firstly, PVQ is a vector quantization method which means it can achieve higher signal-to-noise ratios than scalar quantization methods that quantize weights independently. Secondly, **PVQ uses an implicit codebook**, which means that it does not require an explicit codebook to be constructed in memory. This makes the approach more memory efficient, but more importantly, the implicit codebook size can reach far beyond the memory that would have been required with an explicit codebook. To illustrate, explicitly storing a codebook that quantizes 16 bit precision vectors of a groupsize of 128 to 4 bit per weight would require approximately $2.7 \cdot 10^{154}$ bytes, exceeding the estimated information capacity of the observable universe. With PVQ, we can use implicit codebooks of this size because encoding and decoding are done by an efficient algorithm, not a table lookup. Thirdly, **PVQ is search-free**, which means that vectors can be encoded without having to search over a codebook. This is significant apart from the computational benefits, because it allows for on-the-fly quantization of activations and opens the door to quantization at train time. Lastly, the desired bits per weight after quantization can also be fractions (e.g. $b_{\text{direction}} = 3.5$) and are not limited to integers. The ability to choose the groupsize, and bits for the direction make PVQ highly flexible and can be chosen such that the most optimal trade-off between compression and performance is achieved. We find that PVQ outperforms competitive quantization methods in terms of weight-only and activation quantization.

In this work, our focus is on post-training quantization of weights and activations. Some current works quantize LLMs during training (e.g. Ma et al., 2024). We anticipate that PVQ could be used during training because of the advantages outlined above, though we leave a thorough investigation and comparisons to future work.

## 3.2 SIGNAL-TO-QUANTIZATION-NOISE ON IDEAL GAUSSIAN SOURCE

To assess the theoretical effectivity between different quantization algorithms are, we start by comparing empirical estimates of their performance on an idealized standard Gaussian source, zero mean unit variance. We measure the signal-to-quantization-noise ratio QSNR by averaging the mean squared error between true and quantized signal over 1000 samples in Figure 3. We compare PVQ with E8, a method that has optimal packing of uniformly distributed weights in $D = 8$ dimensions only (Tseng et al., 2024a) and naive rounding-to-nearest (RTN) scalar quantization. We find that PVQ achieves QSNR ratios close to the optimal E8

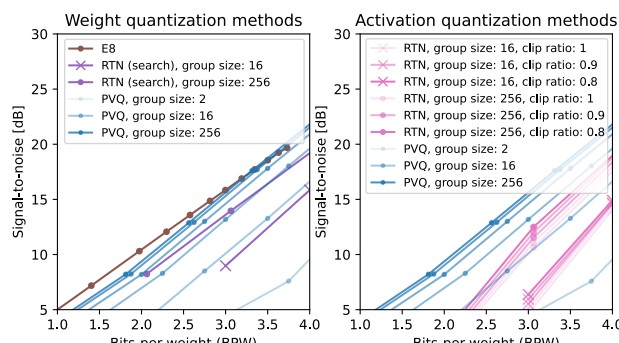

Figure 3: Signal-to-quantization-noise-ratio (QSNR) of quantization methods on standard Gaussian source. PVQ achieves high QSNR close to E8, which uses an optimal packing on a uniform source. PVQ uses an implicit codebook and is search-free, thereby amenable to quantization of weight and activations.

method. Further, PVQ can also be applied to activation quantization, in which case we compare to RTN without search as a baseline that is suitable to quantization of both weights and activations.

## 3.3 PYRAMID VECTOR QUANTIZATION FOR LLMS

This section describes the overall method for quantizing an LLM using PVQ.

**Step 0. Choose the desired BPW.** In PVQ, the trade-off between performance and effective bits per weight (BPW) is controllable through the groupsize $D$, which must divide the number of columns so that $\boldsymbol{W} \in \mathbb{R}^{N \times DG}$, direction bits $b_{\text{direction}} \in \mathbb{N}$ and amplitude bits $b_{\text{amplitude}} \in \mathbb{N}$:

$$\text{BPW} = b_{\text{direction}}/D + b_{\text{amplitude}}/G \tag{7}$$

As the number of direction bits per weight $b_{\text{direction}}/D$, and amplitude bits per weight $b_{\text{amplitude}}/G$ are fractions, it is easy to choose a non-integer number of bits per weight in PVQ.

**Step 1. Coherence processing.** Before we begin quantizing the weights, we perform coherence processing using efficient Hadamard rotation matrices proposed in (Chee et al., 2024), which can be fused into the architecture as described in QuaRot(Ashkboos et al., 2024).

**Step 2. Determine the number of pulses $K$.** To determine the number of pulses $K$ such that the number of bits required to encode PVQ vectors remains within the desired maximum, we find the largest $K$ by increasing it such that it still satisfies $\lceil \log_2(N(D,K))/D \rceil < b_{\text{direction}}$. Here, $N(D,K)$ is computed using the recursive algorithm outlined in Section 2.4. In practice, the number $N(D,K)$ can exceed regular integer types and may require arbitrary precision integers.

**Step 3. Quantizing the direction.** We quantize all LLM weight matrices making up key, query, value and fully-connected components. We write the normalized weight matrix $\boldsymbol{W} = [\boldsymbol{W}_1 \quad \boldsymbol{W}_2 \quad \dots \quad \boldsymbol{W}_G] \in \mathbb{R}^{N \times GD}$, with $N$ features, $G$ groups and a groupsize of $D$. Given our choice of $K$, we quantize groups $\boldsymbol{W}_g \in \mathbb{R}^{N \times D}$ using the quantization procedure of Section 2.4 yielding a quantized direction matrix $\widehat{\boldsymbol{W}}_g$ on the pyramid $\mathcal{P}_{D,K}$, ie. elements are rounded integers and absolute values of rows sum to 1. This operation can be parallelized over features and groups.

**Step 4. Computing the amplitude.** For each quantized row vector $\widehat{\boldsymbol{w}} \in \mathbb{R}^D$ in a group, we can find an optimal rescaling by $s \in \mathbb{R}$ that minimizes the Euclidean distance to the original weight $\boldsymbol{w}$ in closed-form $s = \widehat{\boldsymbol{w}}^T\widehat{\boldsymbol{w}}/\widehat{\boldsymbol{w}}^T\boldsymbol{w}$. Repeating this for all features and groups yields an amplitude matrix $\boldsymbol{S} \in \mathbb{R}^{N \times G}$. We refer to quantized $\widehat{\boldsymbol{W}}$ as the '*direction*' and the scales $\boldsymbol{S}$ as the '*amplitude*'.

**Step 5. Quantizing the amplitude (optional)** For small groupsizes, we can optionally quantize amplitudes as described in Section 3.4. For each row $\boldsymbol{s} \in \mathbb{R}^G$ in $\boldsymbol{S}$, this entails quantizing normalized elements $\lfloor \text{CDF}(s_i^2/||\boldsymbol{s}||_2^2) \cdot 2^b \rfloor$ using the CDF of the Beta$(D/2, D(G-1)/2)$ distribution. The normalizing constant $||\boldsymbol{s}||_2^2$ needs to be stored for dequantization, but because it is shared across groups its contribution to the total $< 0.01$ bits per weight is negligibly small.

**Step 6. Correcting the quantization error.** For each quantized group of weights indexed by the quantized columns $\boldsymbol{s}_g \widehat{\boldsymbol{W}}_g \in \mathbb{R}^{N \times D}$, we can update the other remaining columns $\boldsymbol{W}_{\neg g} \in \mathbb{R}^{N \times (G-1)D}$ to compensate the quantization error that minimizes the proxy loss $L$ of Equation (1):

$$\boldsymbol{W}_{\neg g} \leftarrow \boldsymbol{W}_{\neg g} - \boldsymbol{H}_{\neg g, \neg g}^{-1}\boldsymbol{H}_{\neg g, g}(\boldsymbol{W}_g - \boldsymbol{s}_g \widehat{\boldsymbol{W}}_g) \tag{8}$$

where $\boldsymbol{H}_{\neg g, \neg g}$ is a square submatrix of the inverse Hessian with rows and columns corresponding to remaining weights and $\boldsymbol{H}^{-1}$ is a rectangular submatrix of the inverse Hessian with rows and columns that correspond to quantized and remaining weights respectively. We can avoid inverting the full Hessian for every update by working with its Cholesky decomposition $\boldsymbol{H} = \boldsymbol{L}\boldsymbol{L}^T$, as detailed in GPTQ (Frantar et al., 2022) which proposed the update in the context of scalar quantization.

**Step 7. Encoding the direction.** To encode a quantized weight $\widehat{\boldsymbol{W}} \in \mathbb{R}^{N \times GD}$ (e.g. 32 bit float for $\mathbb{R}$) into the actual low bit integer representation $\overline{\boldsymbol{W}} \in \mathbb{N}^{N \times G}$, sometimes referred to as the 'code', we can use the PVQ encoding algorithm described in Algorithm 2. This is problematic, since the number of bits required to store a vector may exceed the bit width of integer types in many languages (e.g. when $\lceil \log_2(N(D,K)) \rceil > 128$ bits, it can not be represented in a 128 bit integer). To overcome this, we implemented arbitrary precision arithmetic in CUDA/C++ to support arbitrary bit width integer types to allow encoding and decoding kernels.

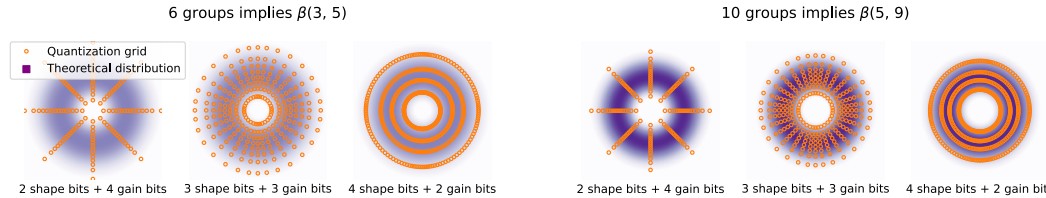

Figure 4: Effect of different direction and amplitude bits on effective PVQ quantization grid. The distribution automatically matches the theoretical Beta weight distribution.

### 3.4 AMPLITUDE QUANTIZATION

**Normalized amplitudes follow the Beta distribution.** Smaller groupsizes lead to a lower quantization error, but lead to an increase in the number of bits required to store amplitude parameters. This is because the amplitude $\boldsymbol{S} \in \mathbb{R}^{N \times G}$ grows linearly with the number of groups $G$, and therefore inversely proportional to the chosen groupsize $D$. To overcome this issue, we propose a theoretically and empirically motivated scheme to quantize amplitude parameters. In Theorem 3.1, we note that row vectors of normalized amplitudes follow a Beta distribution $\text{Beta}(\frac{D}{2}, \frac{D(G-1)}{2})$ of which the coefficients depend on the groupsize $D$ and the number of groups $G$. We empirically confirm that our theory meets practice, as we find that the Beta distribution matches normalized weight distributions of pretrained LLM models, after performing the rotation described in Section 2.3, as shown for selection of layers in a pretrained LLM in Figure 4. To exploit the observation that amplitudes are Beta distributed, we propose to use the quantiles of this distribution to quantize amplitudes.

**Theorem 3.1** *Let $\boldsymbol{w} \in \mathbb{R}^{GD}$ be a normally distributed vector that can be grouped in $G$ equally sized vectors $\boldsymbol{w} = [\boldsymbol{v}_1 \quad \boldsymbol{v}_2 \quad \cdots \quad \boldsymbol{v}_G]$ where each of the vectors $\boldsymbol{v}_g$ has the same dimensionality equal to the groupsize $\boldsymbol{v}_g \in \mathbb{R}^D$. Then the normalized radius (the 'amplitude') of each group $s_g = \boldsymbol{v}_g^T \boldsymbol{v}_g / ||\boldsymbol{w}||_2^2$ follows the $s_g \sim \text{Beta}(\frac{D}{2}, \frac{G(D-1)}{2})$ distribution. (Proof in Appendix A.1)*

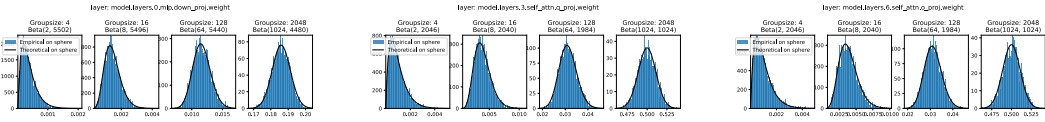

Figure 5: Theoretical Beta distribution of Theorem 3.1 closely match amplitudes of rotated weights in trained LLMs, here demonstrated for empirical weight distributions of a pretrained Llama-v3 8B.

**Quantizing amplitudes using Beta quantiles.** The observation that normalized amplitudes are Beta distributed is important, as it suggests that we can quantize amplitudes efficiently by mapping centers of linearly spaced regions through the quantile function of a Beta distribution, without introducing additional hyper-parameters. To obtain a $b$ bit quantizer, we take a regular uniform grid of $2^b$ points after first transforming elements through a change-of-variables given by the CDF of the Beta distribution and dequantizing using the inverse CDF (see Figure 6):

$$\text{quantize}(x) = \lfloor \text{CDF}(x) \cdot 2^b \rfloor \quad (9)$$

$$\text{dequantize}(x) = \text{PPF}((x + 0.5)/2^b) \quad (10)$$

Figure 6: Quantiles of Beta(2, 6).

where $\lfloor \cdot \rfloor$ denotes the floor function, $\text{CDF}(\cdot)$ the cumulative density function of the Beta distribution and its inverse $\text{PPF}(\cdot)$, known as the percentile point function of the Beta distribution.

# 4 RESULTS

## 4.1 WEIGHT-ONLY QUANTIZATION (WITHOUT AMPLITUDE QUANTIZATION)

Since most prior work focuses on weight-only quantization, we begin by comparing the performance of weight-only PVQ quantization with common weight-only quantization baselines. Following prior work (Frantar et al., 2022; Ashkboos et al., 2024), we measure test perplexity (PPL) and average accuracy on a range of zero-shot downstream tasks after quantizing common open-source LLM models of Phi, Mixtral and Llama families. We compare groupsizes in [16, 32, 64, 128, 256], keep the amplitude in 16 bit, and use direction bits in [3, 3.5, 4, 4.5, 5, 5.5, 6, 7, 8] for PVQ and [3, 4, 5, 6, 7, 8] for other methods. Unlike most scalar quantization methods, PVQ more easily supports non-integer number of direction bits as we can choose an integer number of bits per codeword that is not divisible by the number of groups $G$ – for instance, with groupsize $D = 16$ and 40 bits per group results in $40/16 = 2.5$ direction bits).

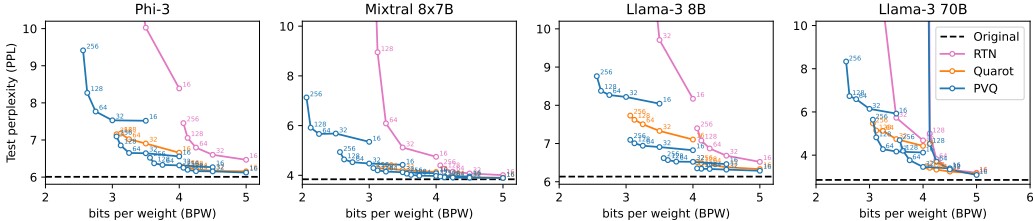

Figure 7: Weight only quantization. Test perplexities (PPL) after quantizing with different methods at group size settings (within connected set) and various direction bits (between connected sets).

|  |  |  |  |  | Phi-3-mini-4k | | Mixtral-8x7B | | Llama-3-8B | | Llama-3-70B | |
|---|---|---|---|---|---|---|---|---|---|---|---|---|
| Method | Hessian | Spherical | Groupsize | BPW | PPL ($\downarrow$) | Avg. Acc ($\uparrow$) | PPL ($\downarrow$) | Avg. Acc ($\uparrow$) | PPL ($\downarrow$) | Avg. Acc ($\uparrow$) | PPL ($\downarrow$) | Avg. Acc ($\uparrow$) |
| Original |  |  |  | 16 | 6.01 | 0.72 | 3.84 | 0.78 | 6.13 | 0.73 | 2.85 | 0.80 |
| RTN |  |  | 128 | 3.125 | 19.03 | 0.53 | 8.95 | 0.66 | 29.41 | 0.41 | 487.94 | 0.45 |
| GPTQ | ✓ |  | 128 | 3.125 | 7.36 | 0.65 | 8.40 | 0.52 | 17.77 | 0.40 | nan* | nan* |
| QuaRot | ✓ | ✓ | 128 | 3.125 | 7.17 | 0.67 | 4.29 | 0.76 | 7.62 | 0.69 | 5.14 | 0.76 |
| PVQ | ✓ | ✓ | 128 | 3.125 | **6.85** | **0.68** | **4.20** | **0.77** | **7.01** | **0.72** | **4.82** | **0.77** |

Table 1: Weight-only quantization in sub-4 bits. Post-quantization test perplexity (PPL) and average zero-shot (Avg. Acc) performance. PVQ yields the highest performance after quantization. Details and additional results in Appendix E.1. nan* indicates GPTQ fails due to non-psd Hessian.

## 4.2 WEIGHT-ONLY QUANTIZATION (DIRECTION AND AMPLITUDE QUANTIZATION)

A benefit of PVQ is that the number of bits for the direction and bits for the amplitudes can be chosen flexibly, even to not-integer ratios. To evaluate the effect of different amplitude bits, we repeat the same experiment as before, but rather than varying the groupsize, we fix the groupsize to 16 and vary the amount of bits used for the amplitude.

|  |  |  |  |  | Phi-3-mini-4k | | Mixtral-8x7B | | Llama-3-8B | | Llama-3-70B | |
|---|---|---|---|---|---|---|---|---|---|---|---|---|
| Method | Groupsize | Hessian | Spherical | BPW | PPL ($\downarrow$) | Avg. Acc ($\uparrow$) | PPL ($\downarrow$) | Avg. Acc ($\uparrow$) | PPL ($\downarrow$) | Avg. Acc ($\uparrow$) | PPL ($\downarrow$) | Avg. Acc ($\uparrow$) |
| Original |  |  |  | 16.00 | 6.01 | 0.72 | 3.84 | 0.78 | 6.13 | 0.73 | 2.85 | 0.80 |
| PVQ [2.5 bit directions, 16 bit amplitudes] | 16 | ✓ | ✓ | 3.50 | **7.52** | **0.67** | **4.42** | **0.76** | **8.04** | **0.68** | **5.92** | **0.74** |
| PVQ [3 bit directions, 4 bit amplitudes] | 16 | ✓ | ✓ | 3.25 | **6.85** | **0.69** | **4.22** | **0.76** | **7.14** | **0.70** | **4.51** | **0.78** |

Table 2: Quantizing direction and amplitude. We compare post-training perplexity (PPL) and average zero-shot performance (Avg. Acc). PVQ yields the highest performance after quantization. Details and additional results in Appendix E.2.

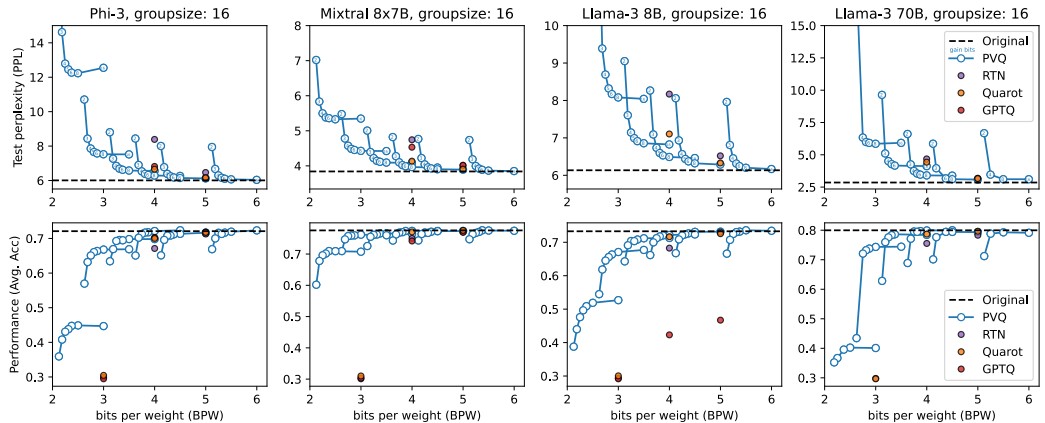

Figure 8: direction and amplitude quantization. Test perplexities (PPL) and test accuracies (Avg. Acc ) at different bits for amplitudes (within connected sets) and various direction bits (between sets).

## 4.3 WEIGHTS AND ACTIVATION QUANTIZATION

As PVQ use an implicit codebook and is search-free, it can be applied to not only the weights but also the activations during inference and reduce computational requirements of the forward-pass. In Figure 13 we evaluate the final test perplexity for different settings of effective bits per weight and bits per activations. In Table 3, we compare the resulting test perplexities to other methods for quantizing weight and activation. We also compare to naive round-to-nearest (RTN) without search, which is also amenable to quantization of weights and activations but has much lower signal to noise. We find that PVQ obtains state-of-the-art performance across the considered LLM architectures, and that this holds generally across different settings of bit rates.

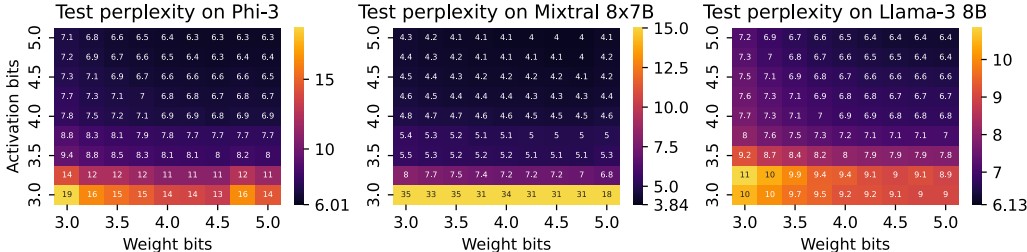

Figure 9: Weights and activations. Comparing test perplexity at different bits per weight and bits per activations. From minimal compression (top right) to high levels of compression (bottom left).

| Method | | | | | | | Phi-3-mini-4k | | Mixtral-8x7B | | Llama-3-8B | |
| Weights | Activations | Groupsize | Hessian | Spherical | BPW | BPA | PPL ($\downarrow$) | Avg. Acc ($\uparrow$) | PPL ($\downarrow$) | Avg. Acc ($\uparrow$) | PPL ($\downarrow$) | Avg. Acc ($\uparrow$) |
|---|---|---|---|---|---|---|---|---|---|---|---|---|
| Original | - | | | | 16 | 16 | 6.01 | 0.72 | 3.84 | 0.78 | 6.13 | 0.73 |
| GPTQ | RTN | 128 | ✓ | | 4.125 | 4.125 | 8.36 | 0.63 | 7.41 | 0.59 | 4747.68 | 0.36 |
| QuaRot | RTN | 128 | ✓ | ✓ | 4.125 | 4.125 | 7.48 | 0.67 | 4.43 | 0.75 | 7.34 | 0.70 |
| PVQ | RTN | 128 | ✓ | ✓ | 4.125 | 4.125 | 7.37 | 0.66 | **4.40** | **0.76** | 7.16 | 0.70 |
| PVQ | PVQ | 128 | ✓ | ✓ | 4.125 | 4.125 | **6.94** | **0.68** | 4.57 | 0.75 | **6.89** | **0.71** |

Table 3: Quantizing weight and activation in 4 bits. We compare perplexity (PPL) and average zero-shot performance (Avg. Acc) after quantizing different open source LLM models using various post-training quantization methods. PVQ yields the highest performance after quantization. Additional results in Appendix E.3.

## 4.4 DOWNSTREAM ZERO-SHOT TASKS

In Table 4, we provide results on downstream zero-shot tasks split out per task. We report weight-only PVQ with both direction and amplitude quantization on a Llama-3-8B model. Additional results including other LLM models can be found in Appendix E.4.

| Method | Groupsize | Hessian | Spherical | BPW | Llama-3-8B PPL ↓ | PQ ↑ | WG ↑ | HS ↑ | A-e ↑ | A-c ↑ | LA ↑ | Avg. ↑ |
|---|---|---|---|---|---|---|---|---|---|---|---|---|
| Original | | | | 16.000 | 6.13 | 0.81 | 0.73 | 0.79 | 0.78 | 0.53 | 0.76 | 0.73 |
| RTN | | | | 3.125 | 29.41 | 0.64 | 0.55 | 0.42 | 0.41 | 0.25 | 0.22 | 0.41 |
| GPTQ | 128 | ✓ | | 3.125 | 17.77 | 0.63 | 0.59 | 0.35 | 0.43 | 0.26 | 0.17 | 0.40 |
| QuaRot | 128 | ✓ | ✓ | 3.125 | 7.62 | 0.77 | 0.71 | 0.73 | 0.75 | 0.46 | 0.71 | 0.69 |
| PVQ [3 bit directions, 16 bit amplitudes] | 128 | ✓ | ✓ | 3.125 | 7.01 | 0.80 | 0.73 | 0.76 | 0.78 | 0.50 | 0.75 | 0.72 |

Table 4: Performance on downstream tasks. We compare performance on zero-shot downstream tasks after quantizing weights using different weight quantization methods.

## 4.5 OPTIMIZING PVQ FOR CUDA

For large $D$ and $K$, PVQ results in large precision integer codes $c$, far surpassing the native 32-bit integer operations on CUDA. As highlighted in Section 3.3, even 128-bit operations introduced with CUDA 11.5 are not sufficient. We therefore implement the CUDA kernels for PVQ on custom subroutines for arbitrary precision integer arithmetic relying on PTX instructions using the *carry-forward* registry (`CC.CF`) for multi-word integer addition, subtraction and bit-shifting. We use a word-minor memory layout to ensure that the memory access can be coalesced. Table 5 presents the time and I/O complexity for our implementations.

For quantization, encoding and decoding we parallelize the work across the batch dimension. Further optimizations are possible, especially for small batch sizes $B$. In particular, the inner loops of each algorithm (Appendix B) can be further parallelized by first sorting or computing the cumulative sum of $|x_i|$ respectively using a CUDA optimized reduction (Harris et al., 2007).

The recurrent formulation for the size table ($N[D, K]$) quickly becomes prohibitive as its naive implementation requires $O(D \cdot K \cdot G)$ serial operations, each requiring overlapping reads and writes. Instead, we reformulate the algorithm to perform $K$ operations in parallel using a CUDA optimized Hillis-Steele type scan reduction (Hillis & Steele, 1986) which accounts for the $\log(K)$ factor in Table 5, computing each row in parallel (Appendix C.2). This has the further advantage of only having to read from global memory for synchronization following (Xiao & Feng, 2010), otherwise the threads keep the values in registry and exclusively writes to global memory.

| **Function** | **Time complexity** | **I/O complexity** |
|---|---|---|
| Quantization | $O(B \cdot D)$ | $O(B \cdot D)$ |
| Encoding / Decoding | $O(B \cdot D \cdot G)$ | $O(B \cdot D \cdot G)$ reads, $O(B \cdot D)$ ordered writes |
| Size table generation | $O(K \cdot \log(K) \cdot D \cdot G)$ | $O(K \cdot D \cdot G)$ ordered writes |

Table 5: The time and I/O complexity on the HBM for our CUDA optimized PVQ kernels, where $B$ is the batch size and $G$ the number of words comprising each integer. All functions have $O(G)$ space complexity. We highlight ordered I/O operations as this more efficient use of the memory bandwidth as they can be coalesced into fewer operations as opposed to random ones.

## 5 CONCLUSION

This work explored pyramid vector quantization (PVQ) for quantization of weights and activations in large language models (LLMs). PVQ is a vector quantization method that allows high signal-to-noise ratios without having to build an explicit codebook or perform search. This results in state-of-the-art quantization performance in terms of the most favourable performance to bits-per-weight trade-off, and is amenable to quantization of activation. This has direct practical benefit for post-training model compression, but also opens the door towards quantization at train time. We propose to quantize LLMs using an implicit PVQ codebook on the unit sphere, which can be flexibly configured for codesize and dimensions. In addition, we propose a theoretically and empirically motivated way to also quantize amplitudes enabling small groupsizes in practice. Lastly, we incorporate Hessian information throughout the process to minimize feature error due to quantization. This yields a novel and highly parallelisable algorithm for LLM weights and activations. We demonstrate state-of-the-art quantization performance in terms of superior performance after quantizing pre-trained models, on both weight-only and weight and activation quantization.

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

# A MATHEMATICAL DETAILS

## A.1 PROOF THAT AMPLITUDES ARE BETA DISTRIBUTED

Assume we have weights ( $w_1, w_2, \ldots, w_N$) that are normally distributed: $w_i \sim \mathcal{N}(0, \sigma^2)$. The sum of the squares of these weights for a subset of size ( D ) is chi-squared distributed $u = w_1^2 + w_2^2 + \ldots + w_G^2 \sim \sigma^2 \chi_B^2$. Now, consider two independent chi-squared distributed variables: $a \sim \sigma^2 \chi_A^2, b \sim \sigma^2 \chi_B^2$. It can be shown that the ratio of these two variables follows a Beta distribution: $\frac{a}{a+b} \sim \text{Beta}\left(\frac{A}{2}, \frac{B}{2}\right)$ (Frankl & Maehara, 1990). This result is independent of the scale parameter ( $\sigma^2$ ). To normalize the weights in a group, we consider the ratio: $\frac{w_1^2 + w_2^2 + \ldots + w_D^2}{w_1 2 + w_2^2 + \ldots + w_N^2}$, where $D < N$. This can be rewritten as: $\frac{w_1^2 + w_2^2 + \ldots + w_D^2}{(w_1 2 + w_2^2 + \ldots + w_D^2) + (w_{D+1}^2 + \ldots + w_N^2)}$. Given that $(w_1^2 + w_2^2 + \ldots + w_D^2) \sim \sigma^2 \chi_D^2$ and $(w_{D+1}^2 + \ldots + w_D^2) \sim \sigma^2 \chi_{N-D}^2$, the ratio follows a Beta distribution: $\frac{w_1^2 + w_2^2 + \ldots + w_B^2}{w_1 2 + w_2^2 + \ldots + w_N^2} \sim \text{Beta}\left(\frac{D}{2}, \frac{N-D}{2}\right)$. Thus, the normalized weights in a group follow a Beta distribution.

# B  Classic PVQ subroutines

The quantization, encoding and decoding algorithms of classic PVQ are provided below. They are equivalent to the algorithms originally proposed in (Fischer, 1986), and included to be self-contained. We also fixed a small bug in the original description of the decoding algorithm.

## B.1  Algorithm 1: PVQ Quantization

---
**Algorithm 1** PVQ quantization: $\boldsymbol{v} \mapsto \widehat{\boldsymbol{v}}, \quad \mathbb{R}^D \to \mathbb{Z}^D$.
---
1: $\widehat{\boldsymbol{v}} \leftarrow \boldsymbol{v}$
2: **while** $||\widehat{\boldsymbol{v}}||_1 \neq K$ **do**
3:    $i \leftarrow \text{argmax}_i(|\boldsymbol{v}_i|)$
4:    $\boldsymbol{v}_i \leftarrow \boldsymbol{v}_i - \text{sign}(\boldsymbol{v}_i)$
5:    $\widehat{\boldsymbol{v}} \leftarrow \text{round}\left(\frac{K}{||\widehat{\boldsymbol{v}}||_1}\widehat{\boldsymbol{v}}\right)$
6: **end while**
7: **return** $\widehat{\boldsymbol{v}}$
---

## B.2  Algorithm 2: PVQ Encoding

---
**Algorithm 2** PVQ encoding: $\boldsymbol{p} \mapsto c, \quad \mathbb{Z}^D \to [1, N(D, K)]$.
---
1: $c \leftarrow 0, i \leftarrow 1, d \leftarrow D, k \leftarrow K$
2: **while** $k! = 0$ **do**
3:    **if** $|x_i| = 1$ **then**
4:        $c \leftarrow c + N(d-1, k) + \frac{1-\text{sgn}(x_i)}{2}N(d-1, k-1)$
5:    **end if**
6:    **if** $|x_i| > 1$ **then**
7:        $c \leftarrow c + N(d-1, k) + 2\sum_{j=1}^{|x_i|-1} N(d-1, k-j) + \frac{1-\text{sgn}(x_i)}{2}N(d-1, k-|x_i|)$
8:    **end if**
9:    $k \leftarrow k - |x_i|$
10:    $d \leftarrow d - 1$
11:    $i \leftarrow i + 1$
12: **end while**
13: **return** $c$
---

### B.3 Algorithm 3: PVQ Decoding

---

**Algorithm 3** PVQ decoding: $c \mapsto \boldsymbol{p}, \quad [1, N(D, K)] \to \mathbb{Z}^D$.

---

1: $\boldsymbol{x} = \boldsymbol{0}$
2: $z \leftarrow 0, i \leftarrow 1, d \leftarrow D, k \leftarrow K$
3: **while** $k > 0$ **do**
4:     **if** $c = z$ **then**
5:         $x_i \leftarrow 0$
6:         **if** $k > 0$ **then**
7:             $x_D \leftarrow k - |x_i|$
8:             $k \leftarrow 0$
9:         **end if**
10:     **else**
11:         **if** $c < cb + N(d - 1, k)$ **then**
12:             $x_i \leftarrow 0$
13:         **else**
14:             $z = z + N(d - 1, k)$
15:             $j \leftarrow 1$
16:             **while** $c \geq z + 2N(d - 1, k - j)$ **do**
17:                 $z \leftarrow z + 2N(d - 1, k - j)$
18:                 $j \leftarrow j + 1$
19:             **end while**
20:             **if** $c < z + N(d - 1, k - j)$ **then**
21:                 $x_i \leftarrow j$
22:             **else**
23:                 $x_i \leftarrow -j$
24:                 $z + N(d - 1, k - j)$
25:             **end if**
26:         **end if**
27:     **end if**
28: **end while**
29: **return** $\boldsymbol{x}$

---

The decode algorithm Algorithm 3 is akin to that described in the original PVQ paper (Fischer, 1986) and provided to be self-contained. It also fixes a missing line 23, setting $x_i \leftarrow -j$. Not including this line results in wrongly decoded vectors when $\boldsymbol{p}$ contains negative values, except for when the last value is negative (which is why the example provided in the original paper does not fail).

## C Implementation details

### C.1 Dataset

We ran all methods on exactly the same data to ensure fair comparison. We follow the Quarot paper (Ashkboos et al., 2024), and use the same 128 samples of the WikiText-2 dataset and hold-out validation data in all experiments.

### C.2 Parallelization using CUDA kernels

Here, we provide additional details for the CUDA implementation described in Section 4.5.

The quantization, encoding and decoding operations described in Appendix B can all be parallelized across the batch dimension. To utilize this, we implementing these operations in custom CUDA kernels. To achieve the time complexities of Table 5 we must re-write the summation in the encoding operation (Algorithm 2, line 7). This is done by in addition to the size table $N$ computing an additional cumulative sum

$$V(d, k) = \sum_{i=1}^{k} N(d, i) \quad \forall d = 0, \ldots, D \text{ and } k = 0, \ldots, K. \tag{11}$$

Using V, the expression becomes

$$\begin{aligned}
c \leftarrow c &+ N(d-1, k) \\
&+ 2\left(V(d-1, k-1) - V(d-1, k - |x_i|)\right) \\
&+ \frac{1 - \text{sgn}(x_i)}{2} N(d-1, k - |x_i|),
\end{aligned}$$

which has constant time and I/O complexity.

Furthermore, the size table function can be parallelized across the $K$ axis, by recognizing that the contributions $N(D-1, K) + N(D, K-1)$ to $N(D, K)$ from Equation 5 correspond to adding the cumulative sum $V(D-1, k)$ to each $k$ at row $D$ which may be done efficiently in CUDA (Harris et al., 2007). The pseudocode for each thread is shown in 4. Notice how the threads only write to global memory, except for any synchornization reads.

---

**Algorithm 4** Size table computation.

---
1: $N(D, K) = \mathbf{0}$
2: $d \leftarrow 0$
3: $k \leftarrow$ thread index
4: $v_k \leftarrow 2$          $\triangleright$ Value in thread registry
5: **while** $d < D$ **do**
6:     **if** $k = 0$ **then**
7:         $v \leftarrow d << 1$          $\triangleright$ Recurrent relationship does not hold for $k = 0$, set to $2d$
8:     **else**
9:         $v_k \leftarrow v_k + v_{k-1}$          $\triangleright$ Add diagonal relationships through warp shuffle
10:     **end if**
11:     $v_k \leftarrow \sum_{i=1}^{k} v_i$          $\triangleright$ Add cumulative sums through a parallelized scan
12:     $N(d, k) \leftarrow v_k$          $\triangleright$ Write to size table in global memory
13:     $d \leftarrow d + 1$
14: **end while**
15: **return** $N$

---

### C.3 AMPLITUDE QUANTIZATION

For scale quantization, we pre-compute a list of 10000 points using `scipy.stats.beta.cdf` onto GPU. We then directly index neighbouring points on this list which we interpolate linearly to obtain quantized values. To dequantize, we use `scipy.stats.beta.ppf` to construct an explicit codebook on GPU, which we can index in parallel to perform dequantization.

# D EMPIRICAL AMPLITUDE HISTOGRAMS

To assess how well our theory of theoretical Section 3.4 matches the empircial weight distribuitons of pretrained LLM models, we compare the empirical weight histograms of all layers of a pretrained LLM model with the expected Beta $\left(\frac{D}{2}, \frac{D(G-1)}{2}\right)$ distribution. We consider the weights of a pretrained Llama-v2-7b after coherence processing and provide the histograms of all weight matrices in the model below. We find that the Beta distribution closely matches the empirical weight distributions in practice.

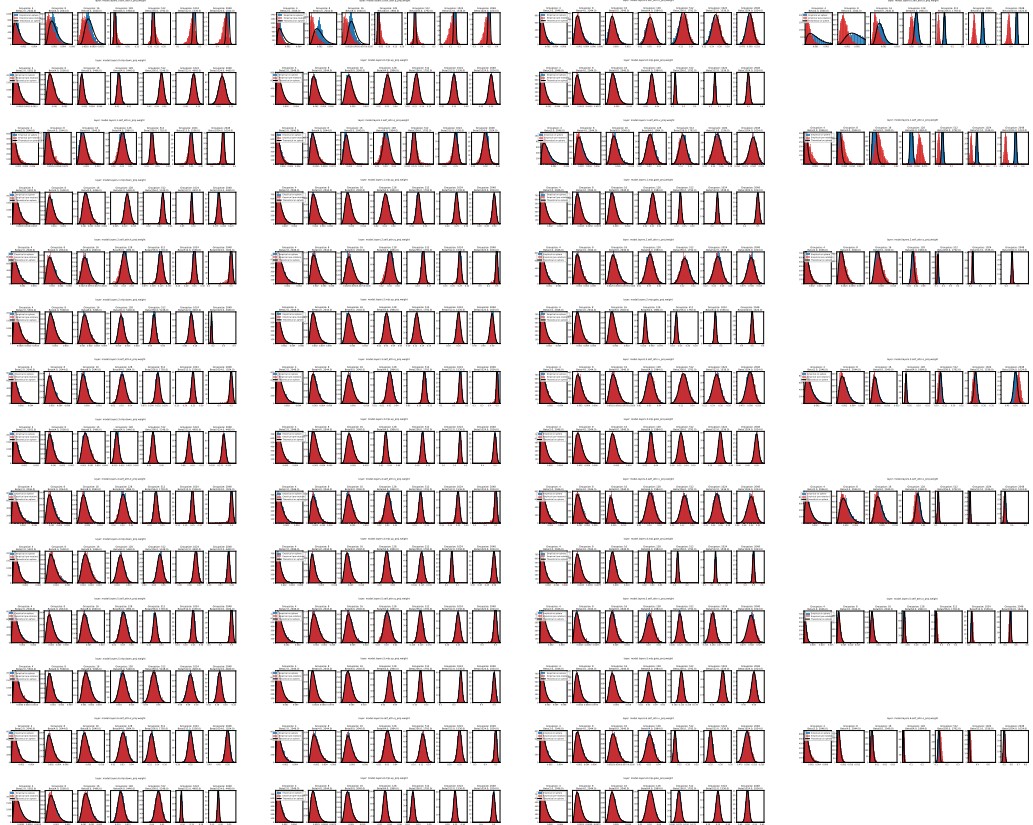

# E  ADDITIONAL RESULTS

## E.1  ADDITIONAL WEIGHT-ONLY EXPERIMENTS (DIRECTION ONLY)

| Method | Groupsize | Hessian | Spherical | BPW | PPL ↑ | PQ ↑ | WG ↑ | HS ↑ | A-e ↑ | A-c ↑ | LA ↑ | Avg. ↑ |
|---|---|---|---|---|---|---|---|---|---|---|---|---|
| | | | | | | | | Phi-3 | | | | |
| Original | | | | 16.000 | 6.01 | 0.81 | 0.73 | 0.78 | 0.79 | 0.57 | 0.65 | 0.72 |
| RTN | | | | 3.125 | 19.03 | 0.72 | 0.58 | 0.62 | 0.60 | 0.41 | 0.25 | 0.53 |
| GPTQ | 128 | ✓ | | 3.125 | 7.36 | 0.77 | 0.66 | 0.69 | 0.74 | 0.51 | 0.53 | 0.65 |
| QuaRot | 128 | ✓ | ✓ | 3.125 | 7.17 | 0.77 | 0.69 | 0.71 | 0.74 | 0.50 | 0.62 | 0.67 |
| PVQ [3 bit directions, 16 bit amplitudes] | 128 | ✓ | ✓ | 3.125 | 6.85 | 0.79 | 0.71 | 0.72 | 0.76 | 0.51 | 0.62 | 0.68 |

Table 6: Performance on downstream tasks. We compare performance on zero-shot downstream tasks after quantizing weights using different weight quantization methods.

| Method | Groupsize | Hessian | Spherical | BPW | PPL ↑ | PQ ↑ | WG ↑ | HS ↑ | A-e ↑ | A-c ↑ | LA ↑ | Avg. ↑ |
|---|---|---|---|---|---|---|---|---|---|---|---|---|
| | | | | | | | | Mixtral 8x7B | | | | |
| Original | | | | 16.000 | 3.84 | 0.84 | 0.76 | 0.84 | 0.83 | 0.60 | 0.78 | 0.78 |
| RTN | | | | 3.125 | 8.95 | 0.78 | 0.66 | 0.69 | 0.71 | 0.45 | 0.64 | 0.66 |
| GPTQ | 128 | ✓ | | 3.125 | 8.40 | 0.69 | 0.60 | 0.47 | 0.47 | 0.30 | 0.60 | 0.52 |
| QuaRot | 128 | ✓ | ✓ | 3.125 | 4.29 | 0.82 | 0.76 | 0.83 | 0.82 | 0.58 | 0.78 | 0.76 |
| PVQ [3 bit directions, 16 bit amplitudes] | 128 | ✓ | ✓ | 3.125 | 4.20 | 0.83 | 0.76 | 0.82 | 0.81 | 0.58 | 0.79 | 0.77 |

Table 7: Performance on downstream tasks. We compare performance on zero-shot downstream tasks after quantizing weights using different weight quantization methods.

| Method | Groupsize | Hessian | Spherical | BPW | PPL ↑ | PQ ↑ | WG ↑ | HS ↑ | A-e ↑ | A-c ↑ | LA ↑ | Avg. ↑ |
|---|---|---|---|---|---|---|---|---|---|---|---|---|
| | | | | | | | | Llama-3-8B | | | | |
| Original | | | | 16.000 | 6.13 | 0.81 | 0.73 | 0.79 | 0.78 | 0.53 | 0.76 | 0.73 |
| RTN | | | | 3.125 | 29.41 | 0.64 | 0.55 | 0.42 | 0.41 | 0.25 | 0.22 | 0.41 |
| GPTQ | 128 | ✓ | | 3.125 | 17.77 | 0.63 | 0.59 | 0.35 | 0.43 | 0.26 | 0.17 | 0.40 |
| QuaRot | 128 | ✓ | ✓ | 3.125 | 7.62 | 0.77 | 0.71 | 0.73 | 0.75 | 0.46 | 0.71 | 0.69 |
| PVQ [3 bit directions, 16 bit amplitudes] | 128 | ✓ | ✓ | 3.125 | 7.01 | 0.80 | 0.73 | 0.76 | 0.78 | 0.50 | 0.75 | 0.72 |

Table 8: Performance on downstream tasks. We compare performance on zero-shot downstream tasks after quantizing weights using different weight quantization methods.

## E.2  ADDITIONAL QUANTIZING DIRECTION AND AMPLITUDE EXPERIMENTS

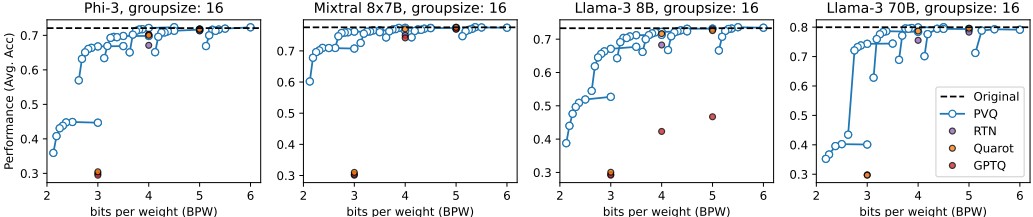

Figure 10: direction and amplitude quantization. Test perplexities (PPL) with different quantization methods at various bit levels for amplitudes (within connected sets) and for direction bits (between connected sets).

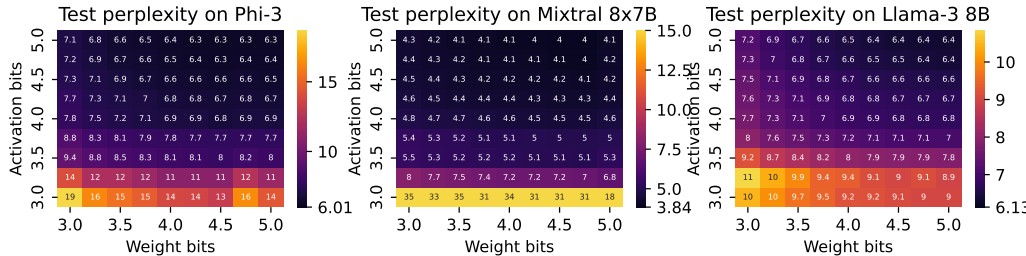

Figure 11: direction and amplitude quantization. Average accuracies (Avg. Acc.) after quantizing with different methods at different bits for amplitudes (within connected set) and various direction bits (between connected sets).

### E.3 ADDITIONAL WEIGHT AND ACTIVATIONS EXPERIMENTS

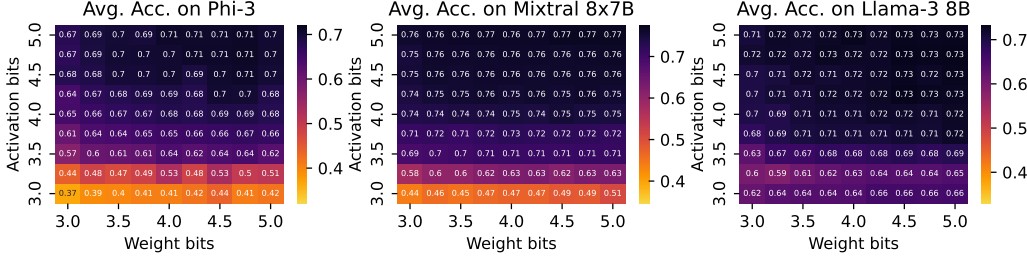

Figure 12: Weights and activations. Comparing test perplexity at different bits per weight and bits per activations. From minimal compression (top right) to high levels of compression (bottom left).

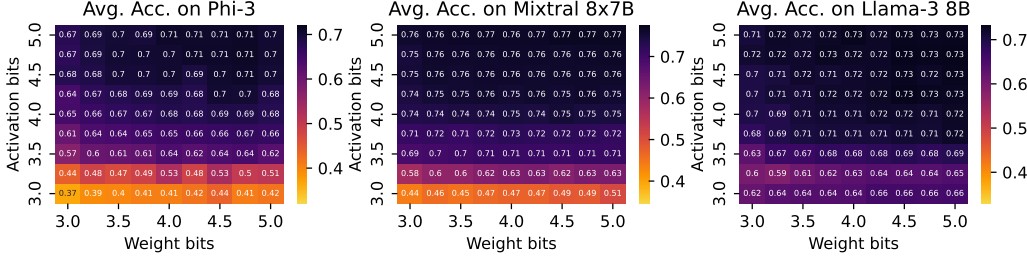

Figure 13: Weights and activations. Comparing average accuracy at different bits per weight and bits per activations. From minimal compression (top right) to high levels of compression (bottom left).

## E.4 ADDITIONAL ZERO-SHOT DOWNSTREAM TASK EXPERIMENTS

| Method | Groupsize | Hessian | Spherical | BPW | PPL ↑ | PQ ↑ | WG ↑ | HS ↑ | A-e ↑ | A-c ↑ | LA ↑ | Avg. ↑ |
|--------|-----------|---------|-----------|-----|-------|------|------|------|-------|-------|------|--------|
| | | | | | | | | Phi-3 | | | | |
| Original | | | | 16.00 | 6.01 | 0.81 | 0.73 | 0.78 | 0.79 | 0.57 | 0.65 | 0.72 |
| RTN | | | | 4.00 | 8.39 | 0.77 | 0.68 | 0.74 | 0.76 | 0.53 | 0.55 | 0.67 |
| GPTQ | 16 | ✓ | | 4.00 | 6.82 | 0.79 | 0.71 | 0.74 | 0.79 | 0.57 | 0.63 | 0.70 |
| QuaRot | 16 | ✓ | ✓ | 4.00 | 6.66 | 0.78 | 0.72 | 0.73 | 0.78 | 0.55 | 0.64 | 0.70 |
| PVQ [3.0 bit directions, 4 bit amplitudes] | 16 | ✓ | ✓ | 3.25 | 6.85 | 0.79 | 0.72 | 0.74 | 0.77 | 0.54 | 0.60 | 0.69 |
| PVQ [3.5 bit directions, 6 bit amplitudes] | 16 | ✓ | ✓ | **3.88** | **6.33** | **0.81** | **0.72** | **0.76** | **0.80** | **0.57** | **0.64** | **0.72** |

Table 9: Performance on downstream tasks after quantizing different open source LLM models using various post-training quantization methods. PVQ yields the highest performance after quantization.

| Method | Groupsize | Hessian | Spherical | BPW | PPL ↑ | PQ ↑ | WG ↑ | HS ↑ | A-e ↑ | A-c ↑ | LA ↑ | Avg. ↑ |
|--------|-----------|---------|-----------|-----|-------|------|------|------|-------|-------|------|--------|
| | | | | | | | | Mixtral 8x7B | | | | |
| Mixtral 8x7B Original | | | | 16.00 | 3.84 | 0.84 | 0.76 | 0.84 | 0.83 | 0.60 | 0.78 | 0.78 |
| RTN | | | | 4.00 | 4.74 | 0.83 | 0.76 | 0.81 | 0.80 | 0.55 | 0.75 | 0.75 |
| GPTQ | 16 | ✓ | | 4.00 | 4.53 | 0.81 | 0.76 | 0.78 | 0.79 | 0.55 | 0.76 | 0.74 |
| QuaRot | 16 | ✓ | ✓ | 4.00 | 4.13 | 0.83 | 0.76 | 0.83 | 0.83 | 0.59 | 0.79 | 0.77 |
| PVQ [3.0 bit directions, 4 bit amplitudes] | 16 | ✓ | ✓ | 3.25 | 4.22 | 0.83 | 0.76 | 0.83 | 0.81 | 0.57 | 0.78 | 0.76 |
| PVQ [3.5 bit directions, 6 bit amplitudes] | 16 | ✓ | ✓ | **3.88** | **3.99** | **0.84** | **0.76** | **0.84** | **0.82** | **0.60** | **0.78** | **0.77** |

Table 10: Performance on downstream tasks after quantizing different open source LLM models using various post-training quantization methods. PVQ yields the highest performance after quantization.

| Method | Groupsize | Hessian | Spherical | BPW | PPL ↑ | PQ ↑ | WG ↑ | HS ↑ | A-e ↑ | A-c ↑ | LA ↑ | Avg. ↑ |
|--------|-----------|---------|-----------|-----|-------|------|------|------|-------|-------|------|--------|
| | | | | | | | | Llama-3-8B | | | | |
| Original | | | | 16.00 | 6.13 | 0.81 | 0.73 | 0.79 | 0.78 | 0.53 | 0.76 | 0.73 |
| RTN | | | | 4.00 | 8.17 | 0.78 | 0.71 | 0.74 | 0.70 | 0.45 | 0.70 | 0.68 |
| GPTQ | 16 | ✓ | | 4.00 | 1415.44 | 0.66 | 0.58 | 0.36 | 0.57 | 0.34 | 0.03 | 0.42 |
| QuaRot | 16 | ✓ | ✓ | 4.00 | 7.10 | 0.79 | 0.73 | 0.76 | 0.76 | 0.51 | 0.75 | 0.72 |
| PVQ [3.0 bit directions, 4 bit amplitudes] | 16 | ✓ | ✓ | 3.25 | 7.14 | 0.78 | 0.72 | 0.76 | 0.74 | 0.49 | 0.74 | 0.70 |
| PVQ [3.5 bit directions, 6 bit amplitudes] | 16 | ✓ | ✓ | **3.88** | **6.53** | **0.80** | **0.73** | **0.77** | **0.76** | **0.51** | **0.75** | **0.72** |

Table 11: Performance on downstream tasks. We compare after quantizing different open source LLM models using various post-training quantization methods. PVQ yields the highest performance after quantization.

