# OpenReview forum: "Pyramid Vector Quantization for LLMs"
_ICLR.cc/2025/Conference — Submitted to ICLR 2025_

### Official Review · Reviewer_T3cg · 2024-10-16

**Soundness:** 3
**Presentation:** 2
**Contribution:** 2
**Rating:** 5
**Confidence:** 5

**Summary:**

In this paper, the authors propose a new post-training quantization algorithm for the quantization of LLMs.
The key idea is to apply a classic pyramid vector quantization (PVQ).
To validate the efficacy of the proposed method, the authors conduct experiments and provide kernels that allow hardware-accelerated encoding and decoding of PVQ.

**Strengths:**

A classic pyramid vector quantization (PVQ) method is well-combined with conventional tricks (e.g., coherence processing, quantization error compensation, Hessian) for the quantization of LLMs.

**Weaknesses:**

1. The authors need to improve the presentation of this paper. It seems that the authors just employed classic encoding and decoding algorithms in the pyramid vector quantization (PVQ) field. Although it is true, the authors need to explain the intuition and meaning of those algorithms. In the current version, the authors just mentioned that they used classic PVQ algorithms, and their pseudocode is provided in the Appendix. Furthermore, the detailed experimental setup is missing. How did the authors construct a calibration dataset? How many data samples are used for the quantization? What is the test dataset?

2. The authors claimed that one key benefit of PVQ is that decoding is search-free while conventional vector quantization algorithms rely on look-up-table-based search. However, this claim is not justified. Why is this a benefit? Is the PVQ decoding process faster than look-up-table-based decoding? Also, while the authors analyze the time complexity of PVQ encoding/decoding in Table 5, actual processing time has not been measured, and thus the reviewer cannot judge whether the proposed algorithm is efficient.

3. The most crucial concern is that while the authors consider vector quantization, all the compared algorithms (GPTQ and QuaRot) are uniform quantization methods. It is natural that vector quantization methods exhibit better performance, but uniform quantization methods are much more hardware-friendly. The authors need to compare the proposed method with existing **vector quantization** methods [1, 2, 3, 4, 5].

[1] GPTVQ: The Blessing of Dimensionality for LLM Quantization, arXiv 2024

[2] Extreme Compression of Large Language Models via Additive Quantization, ICML 2024

[3] QuIP#: Even Better LLM Quantization with Hadamard Incoherence and Lattice Codebooks, ICML 2024

[4] QTIP: Quantization with Trellises and Incoherence Processing, arXiv 2024

[5] VPTQ: Extreme Low-bit Vector Post-Training Quantization for Large Language Models, arXiv 2024


---‐--‐-------------------------
It might be my final chance to adjust the score.

While the authors have mentioned that they achieved similar performance with state-of-the-art methods, I cannot see any results regarding this claim.

Please upload the results about apple-to-apple comparison.

If the authors have uploaded results, I would kindly ask AC to consider my score as 6.

If not, my score is fixed to 5.

Also, while I agree some feedbacks of Reviewer 1xev, I honestly think some comments are very tough to address because the major topic is deep learning, not coding theory.

I hope AC to consider my opinions and other reviewers' comments.

**Questions:**

See the Weaknesses.

---

> ### Author Response · Authors · 2024-11-19
>
> Thank you for your feedback and help to improve the paper. We are glad the reviewers appreciate our use of PVQ in combination with existing approaches commonly used in quantization literature.
>
> > W1a. Intuition behind PVQ
>
> An intuition behind the effectiveness of PVQ is the fact that most important information is captured in the angular information of weight vectors. See also additional reference [4] Kipnis, Alon et al. mentioned by reviewer BTWr.
>
> Further, perhaps counterintuitively, vector quantization with a finite number of quantization points can even result in lower quantization error than scalar quantization in cases where the source is fully independent (e.g., i.i.d. Gaussian data). According to the rate-distortion theory, vector quantization becomes asymptotically more efficient as the number of dimensions D increases. This can be seen geometrically (e.g. Fig 1 in [1]), and we refer to [6, 7] for a more rigorous treatment.
>
> In addition to the high theoretical performance, PVQ comes with a number of practical benefits that make it particularly efficient. Specifically, the fact that PVQ is codebook-free and search-free. Being codebook-free allows very high signal-to-noise ratios and being search-free provides an avenue to develop methods that can be used on not only weights but also activations during inference, ie. no need to run k-means during inference.
>
> > W1b. Used dataset
>
> We thank the reviewer for pointing this out. The experiments follow prior work and use the same datasets for calibration, evaluation and choice of downstream tasks. We ran all methods on exactly the same data to ensure fair comparison. Following the Quarot paper, we use 128 samples of the WikiText-2 dataset and hold-out validation data. We intend to release code upon acceptance with exact run scripts for each experiment, and have included these details in the appendix of the paper to be self-contained.
>
> > W2a. Necessity of an implicit codebook
>
> We thank the reviewer for raising this. The straightforward answer is that an explicit codebook would simply be too large to fit in memory (see example about number of bytes required to store codebook in Sec. 3.1). As a result, the codebook becomes too large to search or even be represented in under any practically feasible memory constraint. For this reason, naive vector quantization methods can only be used with very small groupsizes (D) or using a very small number of quantization points in the codebook. Instead, PVQ uses an implicit codebook which can be effectively used in high dimensions (larger group sizes) and a large number of effective quantization points, since they never have to be constructed explicitly.
>
> > W2b. Efficiency of the algorithm
>
> We thank the reviewer for raising this point. Indeed, having an algorithm alone is not sufficient as this needs to run in a computational efficient manner, which we have now made more clear in the text. We put considerable effort in making sure the encoding/decoding is fast, both algorithmically and on the engineering side, as described in Sec. 4.3. This is highly non-trivial and relies on custom low-level integer arithmetic to be implemented in C. Together, these efforts have allowed fast custom CUDA kernels for quantization, encoding and decoding, which we intend to release upon acceptance. Compared to other methods, such as Quarot, the quantization and encoding does not lead to significant overhead (less than 30% increase in time) for weight-only quantization. We already provide computational complexities in Table. 5 but will include numerical benchmarks in the final version.
>
> > W3. Other vector quantization methods
>
> Some recent works [1-5] have proposed vector quantization methods in the space of LLMs. However, most of these methods [1, 2, 3, 5] rely on explicit codebooks which means they can only practically be scaled to very small group sizes or small codebook sizes. As far we know, only scalar quantization methods can be competitive in the regime and are therefore used in practice. Unlike the vector quantization methods, PVQ can be scaled to such cases. For this reason, our comparison is limited to scalar quantization methods. We agree that it would be interesting to also compare to QTIP [4]. We have made sure that this and other methods are mentioned in the related work.
>
>
> [1] GPTVQ: The Blessing of Dimensionality for LLM Quantization, arXiv 2024
>
> [2] Extreme Compression of Large Language Models via Additive Quantization, ICML 2024
>
> [3] QuIP#: Even Better LLM Quantization with Hadamard Incoherence and Lattice Codebooks, ICML 2024
>
> [4] QTIP: Quantization with Trellises and Incoherence Processing, arXiv 2024
>
> [5] VPTQ: Extreme Low-bit Vector Post-Training Quantization for Large Language Models, arXiv 2024
>
> [6] Gersho, A., & Gray, R. M. 1991. Vector Quantization and Signal Compression. Chapter 8
>
> [7] Cover, T. M., & Thomas, J. A. 2006. Elements of Information Theory, 2nd Edition. Wiley, Chapter 10 on rate-distortion theory.

---

> > ### Comment · Reviewer_T3cg · 2024-11-19
> >
> > Thanks to the authors for their efforts to address my concerns.
> >
> > The authors have replied that they did not compare the performance with other vector quantization methods because those methods can only be scaled to very small group sizes or small codebook sizes.
> >
> > Regarding this response, my question is "Why did not the authors compare the performance for small group sizes or small codebook sizes?"
> >
> > Since the main topic of this work is non-uniform quantization, I still insist that comparison with existing vector quantization methods is necessary, without which I cannot judge the effectiveness of the proposed method.
> >
> > I kindly ask the authors to address this concern first.
> > After resolving this, I will reply to other responses.

---

> ### Author Response · Authors · 2024-11-24
>
> We regret not including a comparison to Trellis-based quantization methods in the paper, such as Quip# [2] and QTIP [3]. The primary reason for this omission is that the models used in these papers and their respective codebases differ slightly from ours (instruction-tuned Llama versus regular Llama, as used in our work). Additionally, the Quip# and QTIP codebases rely on the larger RedPajama dataset, whereas we use the smaller Wikitext-2 dataset, which is more commonly used in compression literature (e.g., GPTQ, Quarot, The LLM Surgeon, etc.). With these significant differences in mind, we have composed a table summarizing the union of downstream tasks reported in both papers.
>
> | Llama-3.1 8B                 |       |       |       |       |       |         |
> | ---------------------------- | ----- | ----- | ----- | ----- | ----- | ------- |
> | Method                       | BPW   | ArcC  | ArcE  | PiQA  | Wino  | AVG     |
> | BF16 (instruct)              | 16.00 | 0.502 | 0.801 | 0.797 | 0.729 | 0.70725 |
> | QuIP# (instruct+large calib) | 4.00  | 0.502 | 0.797 | 0.797 | 0.731 | 0.70675 |
> | QTIP (instruct+large calib)  | 4.00  | 0.502 | 0.796 | 0.794 | 0.734 | 0.7065  |
> | QuIP# (instruct+large calib) | 3.00  | 0.464 | 0.774 | 0.779 | 0.729 | 0.6865  |
> | QTIP (instruct+large calib)  | 3.00  | 0.492 | 0.793 | 0.792 | 0.745 | 0.7055  |
> |                              |       |       |       |       |       |         |
> | BF16 (regular)               | 16.00 | 0.53  | 0.78  | 0.81  | 0.73  | 0.7125  |
> | PVQ (regular+wikitext calib) | 3.88  | 0.51  | 0.76  | 0.8   | 0.73  | 0.7     |
> | PVQ (regular+wikitext calib) | 3.25  | 0.49  | 0.74  | 0.78  | 0.72  | 0.6825  |
>
> To conclusively determine which approach yields strictly better weight-only performance, further experiments would be necessary. However, we believe that these results are already sufficient to conclude that both methods can achieve roughly comparable quantization performance at 3–4 bits for weight-only scenarios. Importantly, [2,3] do not consider weight and activation quantization, which is a significant motivation behind our PVQ approach. We hope the reviewers find the above satisfactory and that the absence of a more thorough investigation into the performance of [2,3] does not negatively impact the recommendation for acceptance.
>
> [1] van Baalen, Mart, et al. "Gptvq: The blessing of dimensionality for llm quantization." (2024).
>
> [2] Tseng, Albert, et al. "Quip#: Even better LLM quantization with hadamard incoherence and lattice codebooks." arXiv preprint arXiv:2402.04396 (2024).
>
> [3] Tseng, Albert, et al. "QTIP: Quantization with Trellises and Incoherence Processing." arXiv preprint arXiv:2406.11235 (2024).

---

> > ### Comment · Reviewer_T3cg · 2024-11-25
> > **Thanks for the reply, but I maintain my score.**
> >
> > Thanks to the authors for their efforts to address my concerns.
> >
> > However, without further experiments on the same LLM and same calibration data, my concerns cannot be addressed.
> >
> > **The easiest solution is to quantize the instruction-tuned Llama3.1-8b model via the proposed PVQ using RedPajama as a calibration dataset.**
> >
> > Then, the authors can easily compare the performance of QuIP#, QTIP, and the proposed PVQ **without conducting additional experiments with the conventional methods.**
> >
> > However, it seems that the authors are reluctant to compare their method directly with conventional vector quantization methods.
> >
> > The authors mentioned that they just followed the convention in the compression field, but this claim is also weird.
> >
> > Why did the authors follow the convention in the scalar quantization field (e.g., GPTQ, QuaRot), instead of the convention in the **vector quantization field (e.g., QuIP#, QTIP)**, while proposing a new method for the vector quantization?
> >
> > Due to the above reasons, I cannot recommend the acceptance of this paper and keep insisting my score.

---

> > > ### Author Response · Authors · 2024-11-26
> > >
> > > We thank the reviewer for their thorough feedback and understand that the comparison with QuIP# and QTIP remains a concern. We are working hard to implement and run a solid comparison between our method and QuIP# and QTIP using the same models and calibration sets. While personal circumstances have slowed progress more than initially hoped, we are doing our very best to include this comparison within the rebuttal deadline. We will keep you up to date on our progress throughout this process. Thank you for your patience and understanding.

---

### Official Review · Reviewer_1xev · 2024-10-22

**Soundness:** 2
**Presentation:** 1
**Contribution:** 2
**Rating:** 1
**Confidence:** 5

**Summary:**

The paper proposes a quantization method based on using representative points on a sphere to compress LLM weights and activation. The authors claim that it provides a significant improvement over existing art, and it is "codebook free" and "search free".

**Strengths:**

The idea of using spherical code to compress weight and activation may be worth studying. If it can be shown convincingly that this approach indeed outperforms others, then it can make a worthwhile contribution to the area of LLM quantization/compression.

**Weaknesses:**

I found the paper poorly written, lacking clarity, and missing some important baseline comparisons to be convincing.

1. Quantization can be used for different purposes in LLMs. It is done for compressing LLMs (either for storage and transportation across platforms, or for reducing memory usage by reusing weights), but it can also be done for approximate computation. Depending on the targetted usage scenario, the evaluation needs to be designed more carefully. It was never made clear what scenario this work was aiming at. I assume the authors are targeting LLM compression for storage and transportation across platforms since bits were used in the evaluation. This assumption, however, leads to another issue. In data compression, after quantization, an entropy coding step (e.g. Huffman or LZ) is usually taken. The authors seem to have completely ignored this step. It is not clear to me why this is not considered since it is considered common practice.

2. The claim of codebook-free and search-free are dubious. The authors appear to take the naive lookup table implementation as the literal interpretation of "codebook" and "search". However, a structured set of representation points, like the proposed pyramid vectors, is still a codebook. Some codebooks have more structure than others, for example, the simple scalar-quantizer is very structured, but a randomly generated codebook may be the least structured. Representation points obtained by K-mean are not well-structured, but it can also be forced to take certain structures if needed. "Search-free" is also not an accurate term. Even for a completely unstructured codebook, there are approximation methods to simplify the "search" step, for example, using a sum of component-wise differences as an estimate to prune away most candidates. I found using the claim of advantage ambiguous in terms of these two "codebook-free" and "search-free"

3. The authors did not compare the approach to standard approaches of "lattice quantization", "lattice quantization+entropy coding", and even "scalar quantization" and "scala quantization+entropy coding". Lattice does not require a lookup table, and has efficient enumeration techniques. In Fig. 3, we see that E_8 lattice is already performing better than the proposed approach. However, the authors did not include E_8 and other lattices in all the other comparisons. In fact, A^*_8 may be even better than E_8 in dimension-8. The lattice-based quantization has been known for over 40 years to perform well, and there are very efficient algorithms that avoid "search". See for example: (a) Conway, John, and Neil Sloane. "Fast quantizing and decoding and algorithms for lattice quantizers and codes." IEEE Transactions on Information Theory 28.2 (1982): 227-232. (b) Conway, J., and N. Sloane. "A fast encoding method for lattice codes and quantizers." IEEE Transactions on Information Theory 29.6 (1983): 820-824. I found the lack of comparison with these well-known techniques unacceptable.

4. Another quantization approach is trellis-based quantization, and it is known to also perform well and reduce computation. The authors should consider providing a comparison.

5. The paper is quite poorly written. A few extreme examples are as follows 1) There is still an unremoved comment in equation (8). 2) Page 4, below (6), it should be "at most" T<D steps. 3) RTN appears in many figures in Section 3, but was not introduced until Section 4. 4) equation (1) is incorrectly/inaccurately written. H is already the Hessian with the expectation over x, then the last expression in (1) should not need to take expectation over x.

6. I do not believe the claim of "the only method with non-integer shape bits" holds water. The codebook size is not chosen arbitrarily, since it is determined by N(D,K). Therefore, though the number of bits is not necessarily an integer, the number of codewords can still only take discrete values.

7. The example given in 3.1 is quite ridiculous. In practice, no one would store a quantization table in this manner. Even for an unstructured codebook, only the representative points will be stored, and then the quantization procedure will find one representative point in the codebook through some algorithm, either more direct or searching.

**Questions:**

1. What is the targetted application?
2. Why is the lattice-based approach not compared carefully?
3. Why is the trellis-based approach not compared?
4. Is scalar quantization compared? Is it significantly worse? How about scalar quantization plus entropy coding?

---

> ### Author Response · Authors · 2024-11-19
>
> Thank you for your feedback and help to improve the paper. We are glad the reviewers find the idea of using a spherical code to quantize weights an interesting avenue of research, and deem methods that outperform existing approaches a worthwhile contribution to quantization/compression literature.
>
> > W1+Q1. Targeted application
>
> The two main goals of this paper are 1) reduce the memory footprint of the model, and 2) reduce the compute required during the forward pass.
>
> We demonstrate reduced memory footprint of the model in Secs 4.1, 4.2 and 4.4. At the same time, the techniques open the door to reduce memory footprint of the forward pass and allow simultaneous quantization of both weights and activations, for which we demonstrate effectiveness in Sec. 4.3.
>
> One thing that we did not emphasise well in our paper is the computation using PVQ quantized methods. An attractive aspect of scalar quantization is that it can be used directly in computation (assuming hardware implementations are available, which they are for e.g. int4). By contract, most VQ methods must decompress, and compute in the original dtype (i.e. fp16 in most cases). PVQ compresses into _integer codebooks_, and so strikes an attractive balance in requiring some computation for decompression (from codes to vectors of int) but with compute that can ultimately happen in low bit-widths.
>
> We thank the reviewer for raising this and have made sure our motivations are clear from the main text. We expect the clarification above to resolve the issue, but are happy to answer any further questions.
>
> > W2. Codebook-free and search-free
>
> We thank the reviewers for raising this. We make these comments in the context of other quantization methods. The term codebook-free strictly refers to having to represent the codebook in memory. Indeed, pyramid vectors in our case form a codebook. Yet, we call this codebook ‘implicit’ (codebook-free) since these vectors never have to be represented in memory, ie. an (efficient) algorithm maps weight vectors to the index without having to perform a table lookup. This is the crucial benefit of PVQ (!), and we have made sure this is more clear in the text. In naive vector based quantization using finite clusters found with K-means (either unstructured or made structured), the centroids need to be represented in memory which limits scalability to high dimensions and large codebooks. We agree that we were not specific enough with the term search-free, in particular because we did not discuss approximate search algorithms. We fix this by being more precise in our definition of being search-free, relating to not having to perform an exhaustive lookup that scales at least linearly in algorithmic complexity with the size of the codebook. In addition, we have replaced all references to ‘codebook-free’ by ‘implicit codebook’ (in contrast to an ‘explicit codebook’), to avoid confusion. We expect this to resolve the issue.
>
> > W3+Q2. Comparison with lattice based quantization
>
> Apart from being difficult to implement, the most important reason for not considering E8 and A8 lattice based quantization is that they are limited to 8-dimensions. The restriction to 8 dimensions has a negative effect on the efficiency that can be obtained when implementing the quantization method on low-level tensorcores. Further, it limits the ability to choose an optimal point on the pareto-front for different groupsizes, which can have a large effect on performance as indicated by our experiments. We do agree with reviewers that it would be interesting to more extensively compare to lattice based methods. Yet, taking the above into consideration, we decided to only include E8 in the Gaussian source experiment and not in the bigger LLM experiments in which different group sizes are considered.
>
> > W4+Q3. Trellis-based comparison
>
> We are aware of one other work that uses Trellis-based quantization [1]. We agree that it would be interesting to also compare this method. We have made sure the method is added to related work.
>
> > W5. Typos
>
> We thank the reviewer for pointing these out. Typos have been fixed.

---

> > ### Author Response · Authors · 2024-11-19
> >
> > > W6. Non-integer bits
> >
> > We thank the reviewer for raising this point. Please note that bits per weight measures the amount of bits per individual weight, not the amount of bits per weight vector (bits per group). Indeed, N(D, K) remains an integer but since the method is a vector quantization method the number of bits need to be divided by the number of weights per vector D, which means the number of *bits per weight* can be fraction-valued. To give an example, quantizing sets of 16 weights (groupsize D=16) and setting N(D, K) to an integer such that we have 40 bits per vector would result in 40/16=2.5 bits per weight. Since this is quite specific to vector quantization we are aware that this is easily missed by readers that are most familiar with scalar quantization. We thank the reviewer for attending us about this fact and have made this more clear in Sec. 3.3 and Sec. 4.1.
> >
> > > W7. Size of codebook
> >
> > We are not sure what is meant with the quantization table, but we refer to the codebook as the collection of representative points (e.g. the centroids in naive vector quantization using KNN clustering). We understand why the example may appear exaggerated, as any practical method would restrict the number of points to fit memory constraints (limiting effectiveness) or use an implicit codebook, like PVQ. However, the example is meant to illustrate how quickly the memory of an explicit codebook grows in vector quantization with an increasing number of dimensions. The used numbers represent practical groupsize and bits per width, were one to explicitly store the PVQ codebook like one would in naive vector quantization (e.g. unstructured clustering of centroids). We hope this clarifies any questions and makes our considerations more clear.
> >
> > > Q4. Scalar quantization
> >
> > We _did_ directly compare to scalar-quantization with grouping methods in our work, using the latest methods that are applicable for weight-and-activation quantization (QuaRot with GPTQ/RTN quantization).
> >
> >
> > [1] Tseng, Albert, et al. "QTIP: Quantization with Trellises and Incoherence Processing." arXiv preprint arXiv:2406.11235 (2024).

---

> > > ### Comment · Reviewer_1xev · 2024-11-20
> > >
> > > 1. The authors appear to suggest that model storage is not part of the considered applications, though they did not answer directly whether entropy coding is a possibility. This would restrict the application to a narrower field, thus limiting the impact, but does make the other parts more plausible, as there was no entropy coding.
> > >
> > > 2. My concern regarding explicit/implicit codebooks remains. This essentially is not a significant advantage.
> > >
> > > 3. The concern about the comparison with other well-known lattices remains. Contrary to the authors' statement,  A and A* lattices exist for any dimension n>=2, i.e., A2, A3, A4,..., A*2, A*3, A*4,.... These can be used to compare the performance. Unless there is a clear comparison, the advantage of the proposed method is not very convincing.
> > >
> > > 4. The concern regarding the non-integer remains. Non-integer value itself is not that important, but rather, if the method allows a continuous sweeping of the rates, then it is quite desirable. If say, in the example, any value between 0.24 to 3 per weight can be obtained, then it is more meaningful in practice.
> > >
> > > 5. The trellis-coded quantization technique can be found here: Fischer, T.R., Marcellin, M.W. and Wang, M., 1991. Trellis-coded vector quantization. IEEE Transactions on Information Theory, 37(6), pp.1551-1566. There were also many follow-up works on this.

---

> ### Author Response · Authors · 2024-11-24
>
> -1. In our answer we _do_ consider model storage as part of the considered applications, which is what we meant by "1) reduce the memory footprint of the model". PVQ already describes an algorithm that performs the isomorphic (lossless) mapping between (quantized) points on the integer lattice and a binary representation. We report experiments on the resulting bits per weight in Secs 4.1, 4.2 and 4.4. We hope this clarifies the remaining confusion. If not, could the reviewer be more precise with how they propose to use entropy coding in this context?
>
> -2. Having an implicit codebook is one of the major advantages of PVQ, as the representative points (integer lattice) never need to be stored in memory. While this is standard for some simple quantization schemes (e.g., rounding to the nearest value), it is not generally the case. Specifically, many vector quantization approaches rely on clustering and require centroids or representative points to be stored in memory. For instance, [1] uses clustering and stores centroids, yet it does not demonstrate vector quantization beyond d=4 dimensions. In contrast, our PVQ-based vector quantization approach scales effortlessly to dimensions of d>1000 and supports effective codebook sizes that are orders of magnitude larger (see example in Sec. 3.1) than what could be explicitly stored in memory, without any issue. Could the reviewers clarify why having implicit codebooks is not considered an advantage?
>
> -4. We fully agree. The advantage of non-integer value is not important in itself, but allows for a very dense sweeping of possible rates, which is meaningful in practice.
>
> -3+5. We regret not including a comparison to Trellis-based quantization methods in the paper, such as Quip# [2] and QTIP [3]. The primary reason for this omission is that the models used in these papers and their respective codebases differ slightly from ours (instruction-tuned Llama versus regular Llama, as used in our work). Additionally, the Quip# and QTIP codebases rely on the larger RedPajama dataset, whereas we use the smaller Wikitext-2 dataset, which is more commonly used in compression literature (e.g., GPTQ, Quarot, The LLM Surgeon, etc.). With these significant differences in mind, we have composed a table summarizing the union of downstream tasks reported in both papers.
>
> | Llama-3.1 8B                 |       |       |       |       |       |         |
> | ---------------------------- | ----- | ----- | ----- | ----- | ----- | ------- |
> | Method                       | BPW   | ArcC  | ArcE  | PiQA  | Wino  | AVG     |
> | BF16 (instruct)              | 16.00 | 0.502 | 0.801 | 0.797 | 0.729 | 0.70725 |
> | QuIP# (instruct+large calib) | 4.00  | 0.502 | 0.797 | 0.797 | 0.731 | 0.70675 |
> | QTIP (instruct+large calib)  | 4.00  | 0.502 | 0.796 | 0.794 | 0.734 | 0.7065  |
> | QuIP# (instruct+large calib) | 3.00  | 0.464 | 0.774 | 0.779 | 0.729 | 0.6865  |
> | QTIP (instruct+large calib)  | 3.00  | 0.492 | 0.793 | 0.792 | 0.745 | 0.7055  |
> |                              |       |       |       |       |       |         |
> | BF16 (regular)               | 16.00 | 0.53  | 0.78  | 0.81  | 0.73  | 0.7125  |
> | PVQ (regular+wikitext calib) | 3.88  | 0.51  | 0.76  | 0.8   | 0.73  | 0.7     |
> | PVQ (regular+wikitext calib) | 3.25  | 0.49  | 0.74  | 0.78  | 0.72  | 0.6825  |
>
> To conclusively determine which approach yields strictly better weight-only performance, further experiments would be necessary. However, we believe that these results are already sufficient to conclude that both methods can achieve roughly comparable quantization performance at 3–4 bits for weight-only scenarios. Importantly, [2,3] do not consider weight and activation quantization, which is a significant motivation behind our PVQ approach. We hope the reviewers find the above satisfactory and that the absence of a more thorough investigation into the performance of [2,3] does not negatively impact the recommendation for acceptance.
>
> [1] van Baalen, Mart, et al. "Gptvq: The blessing of dimensionality for llm quantization." (2024).
>
> [2] Tseng, Albert, et al. "Quip#: Even better LLM quantization with hadamard incoherence and lattice codebooks." arXiv preprint arXiv:2402.04396 (2024).
>
> [3] Tseng, Albert, et al. "QTIP: Quantization with Trellises and Incoherence Processing." arXiv preprint arXiv:2406.11235 (2024).

---

> > ### Author Response · Authors · 2024-11-26
> >
> > We thank the reviewers for their thorough feedback and understand that the comparison with QuIP# and QTIP remains a concern. We are working hard to implement and run a solid comparison between our method and QuIP# and QTIP using the same models and calibration sets. While personal circumstances have slowed progress more than initially hoped, we are doing our very best to include this comparison within the rebuttal deadline. We will keep you up to date on our progress throughout this process. Thank you for your patience and understanding.

---

> ### Comment · Reviewer_1xev · 2024-11-29
> **confusing responses from the authors**
>
> I found the responses by the authors very confusing and unhelpful.
>
> The authors appear unaware of the basic difference between fixed-level (fixed-rate) quantization and entropy-constrained quantization, which have been studied for at least 50 years. It is well-known that after quantization, universal lossless data compression (entropy coding) can significantly improve the overall compression performance, at a slight sacrifice in processing speed. Nevertheless, this technique is very important for model storage and porting models across platforms that are not extremely time-sensitive. I asked the entropy coding question directly in the review, but the authors only gave a confusing indirect answer. Many universal lossless data compression algorithm exists in the literature, Huffman, LZ77, LZ78, CTW, BWT, AC, and even gzip can potentially be used. I'm surprised that the authors need to ask how to apply entropy coding after quantization. I tried to make sense of the authors' first replies, but it appears that the authors are not aware of the important difference between quantization for storage and quantization for computation, and insisted that they are aiming also for model storage. In this case, comparisons with methods using entropy coding must be conducted carefully and included.
>
> The authors claimed a "dense sweeping of the rates". However, this was not evaluated. How dense is it? Quantization based on lattices can also have a sweep of the rates, and by scaling the lattice, it can also be "dense". Is the proposed method denser? I highly suspect this is not the case.
>
> I also strongly agree with review T3cg that for a new vector quantization method, careful comparison with well-known vector quantization approaches must be included and done comprehensively. In fact, the reference [2] Tseng, Albert, et al. "Quip#: Even better LLM quantization with hadamard incoherence and lattice codebooks." arXiv preprint arXiv:2402.04396 (2024), is not a sufficient baseline as lattice-based approach as only E8 is used. I had also pointed out there exists much more efficient quantization approaches that were found over 40 years ago.
>
> The authors also appear unaware of the existence of lattices A_n, A_n^* which offer competitive performance for many general dimensions. I pointed this out to the authors in my reply, yet the authors did not respond to this at all. Lattice A and A* lattices exist for all dimensions, and there are simple algorithms to use them for quantization. These lattices do not require storing codebooks at all. Trellis-based quantization also does not require storing the codebooks explicitly. Thus the authors' claim of implicit codebook does not hold water.
>
> In summary, I found the advantages claimed by the authors on sweeping compression rates and implicit codebook both non-existent, and the performance advantage is also not convincing due to the lack of comprehensive comparison with lattice and trellis-based methods. Moreover, the authors' response confuses several key concepts and suggests a lack of awareness of important existing tools in this space. I therefore decided to reduce the score to 1.

---

### Official Review · Reviewer_BTWr · 2024-11-03

**Soundness:** 2
**Presentation:** 2
**Contribution:** 2
**Rating:** 6
**Confidence:** 4

**Summary:**

The paper proposes to use a known vector quantization (VQ) method called Pyramid VQ (PVQ) to compress a large language model (LLM) at the bit level. The goal is to maximize the tradeoff between LLM's performance and the number of bits per weight.

The paper describes the process of applying PVQ to LLMs, provides performance evaluations and comparisons to other VQ methods, and provides a result about the distribution of a normal random vector projection onto a sphere.

**Strengths:**

The topic of quantizing LLM's is timely and important. Quite a few commercial tools offer bit-level compression of LLMs and it appears that theoretical treatment of the topic is lagging.

Analyzing and discussing PVQ in the context of LLMs are solid contributions because the method is used in some applications. The PVQ description and illustrations are clear and engaging.

Performance comparison to other methods is generally relevant although I have comments below.

**Weaknesses:**

The discussion concerning the advantages of PVQ over other quantization techniques is misleading because the advantage of an implicit codebook generally exists in all practical VQ techniques.  See standard references in the topic like:

[1] Gray, Robert M., and David L. Neuhoff. "Quantization." IEEE Transactions on Information Theory 44, no. 6 (1998): 2325-2383.

Additionally, there is no discussion on the motivation for using VQ compared to scalar quantization in LLMs. Here I'd expect the authors to mention some property of the distribution of attention weights (say) like dependency that makes VQ particularly effective. Other competing methods such as RTN, QuaRot, and QGPT mentioned in the paper are also not fully explained and it is unclear what are the benefits of the proposed method and whether they can be truly attributed to PVQ or the detailed implementation. The coherence processing is perhaps attractive from a practical perspective but it ``hides'' structure in the signal that is ideally exploited by the compressor (not to be confused with `dithering' in scalar quantization that has a different effect).

Numerical evaluations are only against three methods. Because encoding and decoding are not straightforward, I believe that you should also compare runtime information. Indeed, one can easily think about an elaborated lossy weight compression procedure that would certainly outperform all methods but spend considerable time for encoding and decoding (again, see discussion in [1]). There is no comparison to the popular scalar quantization methods (such as the one described in https://huggingface.co/docs/bitsandbytes/en/reference/nn/linear4bit), no comparison to the E8 method that the paper references. There is no explanation of what are the zero-shot tasks. PPL values appear small -- perhaps you mean log(PPL)?

In Section 3.2 the authors say that E8 is a ``state-of-the-art'' method that exploits ``geometric structure''. Both terms in parentheses need explanation.

The theoretical result (Theorem 1) appears to be a reformulation of well-known properties of the distribution of projection of Gaussian vectors. I've found via a quick search the reference:
[2] Frankl, Peter, and Hiroshi Maehara. "Some geometric applications of the beta distribution." Annals of the Institute of Statistical Mathematics 42 (1990): 463-474.
which explains the main point and includes references to earlier works.

For the amplitude ('gain'), the authors first argue that it makes sense to use a scalar quantizer optimized for the Beta distribution (Theorem 1). However, the quantizer they describe is obtained via quantile transform of a Beta-distributed random variable and is not optimal in general. See the discussion about optimal scalar quantization form [1] and from
[3] Lloyd, Stuart. "Least squares quantization in PCM." IEEE Transactions on Information Theory 28, no. 2 (1982): 129-137.

In Section 2.5, it is said the encoder maps a vector to its **closest point** on the pyramid, but the mapping procedure to follow does not necessarily lead to the closest point in the Eecleadian sense.

In Section 3.1, the fact that decoding is done by an ``algorithm'' is meaningless unless you can say something about the runtime of this algorithm.


There are several writing issues:
Unclear what is "infeasibly large".
The 0.01 in Step 5 is misplaced.
Eq. (8) has a weird text.
Typo in Theorem 1 (s_h should probably be s_g)
PPF in Eq. (10) is undefined.
I suggest the following notational changes: 'gain' -> 'amplitude', 'shape' -> 'direction', 'SNR' -> 'QSNR' (Read: signal to quantization-noise ratio).

**Questions:**

The background and motivation need improvement. First, I suggest motivating VQ over scalar quantization which is already widely used in LLMs. Next, you can make the case that PVQ is an interesting VQ technique: It can be implemented in high-dimension due to an implicit codebook (like some other VQ techniques) and is also used in some applications. It appears to me that PVQ provides useful encoding under an L1 metric, whereas other methods would be better under, say, the more common L2. Analyzing the performance of PVQ may still be a solid contribution, but the discussion must be more comprehensive.

Please explain more about the E8 method you provided as "state of the art". Can you compare performance to this method?
Please elaborate more on the other competing methods. Could you also show the performance of 4-bit scalar quantization, say using: https://huggingface.co/docs/bitsandbytes/en/reference/nn/linear4bit

It is well known, hence should be mentioned, that in high dimensions the information in the amplitude ('gain') is negligible compared to the information in the direction ('shape'). For example, see
[4] Kipnis, Alon, and Galen Reeves. "Gaussian approximation of quantization error for estimation from compressed data." IEEE Transactions on Information Theory 67, no. 8 (2021): 5562-5579.

The quantizer you used for the vector's magnitude is not optimized for the Beta distribution in any known sense. The authors either implement the optimal quantization method (under, say, mean squared error) or justify their approach.

Concerning Theorem 1, the authors should discuss how their result relates to or builds upon existing works like [2].

**Details Of Ethics Concerns:**

non

---

> ### Author Response · Authors · 2024-11-19
>
> Thank you for your feedback and help to improve the paper. The reviewer appreciated that quantization is an important topic, and discussing PVQ is a solid contribution. Further, we appreciate the reviewer who found illustrations clear and engaging.
>
> > Motivation behind spherical vector codes.
>
> An important motivation behind exploring spherical vector codes has been recent observations in literature that have demonstrated that directional information of LLM weights are uniformly distributed across the sphere. Further, our observation that LLM weights empirically closely align with the predicted Beta distribution contributes additional empirical evidence to this fact. We thank the reviewers for raising this and we have changed the text in Sec. 2.3 such that this motivation is more clearly stated.
>
> > Implicit codebook in practical VQ techniques. [Q1]
>
> We agree with the authors that several other vector quantization exists that utilise an implicit codebook. Yet, there have been recent proposals to use vector quantization in the context of LLM models which do not use implicit notebooks and rely on an explicit notebook instead (e.g. van Baalen, et al.). In our work, we leverage insights of PVQ, a particularly flexible and practical VQ method with implicit codebooks to improve quantization for LLMs. The benefits of implicit codebooks remain, but we agree with the reviewer and have adapted the text such that it is clear that PVQ is not the only VQ method with implicit codebooks and have included Gray, et al. 1998. We expect this to resolve the concern by the reviewer.
>
> > Vector quantization versus scalar quantization [Q1]
>
> Indeed, vector quantization could offer improvements when source data has special structure or the signal is correlated in certain ways. However, and perhaps counterintuitively, vector quantization with a finite quantization grid can even result in lower quantization error than scalar quantization in cases where the source is fully independent (e.g., i.i.d. Gaussian data). According to the rate-distortion theory, vector quantization becomes asymptotically more efficient as the number of dimensions D increases. This can be seen geometrically (e.g. Fig 1 in [3]), and we refer to [1,2] for a more rigorous treatment.
>
> > Numerical performance of encoding/decoding. [Q2]
>
> We thank the reviewer for raising this point. We have put considerable effort in making sure the encoding/decoding is fast, both algorithmically and on the engineering side, as described in Sec. 4.3. This is highly non-trivial and relies on custom low-level integer arithmetic to be implemented in CUDA. Together, these efforts have allowed fast custom CUDA kernels for all quantization, encoding and decoding, which we intend to release upon acceptance. Compared to other methods, such as Quarot, the quantization does not (<+20%) lead to significant overhead for weight-only quantization. We already provide computational complexities in Table. 5 but will include numerical benchmarks in the final version.
>
> > QLoRA
>
> The method that you shared [5] is used in the context of QLoRA. The focus of that method is fine-tuning, which is out of the scope of this work, and the underlying quantization method that is used is scalar quantization (with grouping). We _did_ directly compare with scalar-quantization with grouping methods in our work, using the latest methods that are applicable for weight-and-activation quantization (QuaRot with GPTQ/RTN quantization).
>
> > E8 [Q2]
>
> The reason we included E8 in the Gaussian source experiment is because it is theoretically optimal for uniformly distributed weights. Yet, we chose not to consider this method, which can be hard to implement, in our LLM quantization experiments for several reasons. Firstly, the method has not been proposed with activation quantization in mind, which is a long term goal. Secondly, E8 is restricted to groupsizes of D=8 which limits its flexibility and practical applicability (i.e. effective cuda implementations are impossible due to tensorcore dimensions) . Lastly, while E8 is optimal on uniform densities we are motivated by recent results demonstrating that LLM weights are distributed along the surface of the sphere, as also evidenced by our empirical results. We trimmed down our statements around E8 and do not refer to it anymore as being a ‘state of the art’ in the final version.
>
> > Directional information in high-dimension [Q3]
>
> We thank the reviewers for bringing up [4]. We consider this highly relevant reference that strengthens the motivation behind using PVQ in the context of LLMs. We have added the reference to Sec. 2.3.

---

> > ### Author Response · Authors · 2024-11-19
> >
> > > Beta quantile [Q4]
> >
> > We propose to quantize the magnitudes (gains) using the quantiles of the expected Beta distribution, as described in paragraph “Quantizing gains using Beta quantiles” of Section 3.2. Under this distribution, the optimal (in squared error) finite set of quantization points corresponds to the quantiles of the same distribution. Thus, the described gain/amplitude quantization method is optimal in some sense. We have updated the manuscript to state this more clearly.
> >
> > > Mapping to closest point on pyramid
> >
> > To map points to the pyramid we follow the procedure of the original PVQ paper [6], which does converge to the “closest point” (see Sec. IV of [6]). It is true that this is not necessarily the closest point when also considering scaling effects in conjunction. If this is what is meant by the reviewer, we would like to also point out that we rescale all vectors after Step 3 using a GPTQ-like Hessian update described in Step 4. which finds the closest optimal amplitude parameter (gain) in the Euclidean sense under the fixed choice of direction (shape). To be sure such details can be reproduced correctly we intend to release code upon acceptance, and have also made sure these details are clear in the paper.
> >
> >
> > > Typos.
> >
> > All typos have been fixed. The terms shape/gain are used in prior literature, but we do recognize that this can be confusing. To improve readability, we have replaced the terms by direction/amplitude, as suggested by the reviewer.
> >
> >
> > [1] Gersho, A., & Gray, R. M. (1991). Vector Quantization and Signal Compression.
> >
> > [2] Cover, T. M., & Thomas, J. A. Elements of Information Theory, 2nd, Chapter 10, 2016
> >
> > [3] van Baalen, Mart, et al. "Gptvq: The blessing of dimensionality for llm quantization." (2024).
> >
> > [4] Kipnis, Alon, and Galen Reeves. "Gaussian approximation of quantization error for estimation from compressed data." 2021
> >
> > [5] Dettmers, Tim, et al. "Qlora: Efficient finetuning of quantized llms." NeurIPS 2024
> >
> > [6] Fischer, Thomas. "A pyramid vector quantizer." 1986
> >
> > [7] Frankl, Peter, and Hiroshi Maehara. "Some geometric applications of the beta distribution." 1990

---

> ### Comment · Reviewer_BTWr · 2024-11-24
>
> Thank you for your response.
>
> VQ motivation. I agree that VQ offers a theoretical advantage in terms of bits per weight up to ~1.53 db in variable rate scalar quantization for iid sources. However, efficient quantization is not only about sphere packing but also about **shaping** when the data source is not iid. It is unclear whether your approach efficiently shapes the data -- especially due to coherence processing. It is not surprising that the data is uniformly distributed over the sphere after coherence processing. Therefore, it appears that you need to motivate coherence processing.
>
> VQ vs SQ. I am curious to understand whether theoretical VQ advantage is significant in practice. Can you provide such a comparison (say, using https://huggingface.co/docs/bitsandbytes/en/reference/nn/linear8bit instead of the 4-bit one I originally referred to if you believe that fine-tuning is out of scope) or explain why it is irrelevant? To be clear, I understand that QuaRot with GPTQ/RTN quantization uses scalar quantization but https://huggingface.co/docs/bitsandbytes/en/reference/nn/linear8bit is popular and hence needs attention.
>
> Optimal quantization of the beta distribution. As I mentioned, the quantizer you propose is not optimized for the beta distribution because of the non-linear quantile transformation. Please make sure to fix your quantizer or say that you use a sub-optimal quantizer.
>
> Theorem 1. "The authors should discuss how their result relates to or builds upon existing works like [2]."

---

> > ### Author Response · Authors · 2024-11-24
> >
> > We thank the reviewer for participating in the rebuttal discussion.
> >
> > > VQ motivation.
> >
> > We thank the reviewer for raising this point. We agree that shaping the data source is important. Our work assumes that data is relatively uniformly distributed on the sphere, and we provide some prior work to support this assumption. To further strengthen our empirical analysis and specifically measure the effect of coherence processing, we have extended our analysis of weight distributions to include both the pre- and post-coherence processing states. These additional results can be found in Appendix D (depicted as blue and red histograms). As can be observed, most amplitudes are already Beta-distributed; however, in cases where this is not initially true, they align with our theoretical Beta distribution after coherence processing (a form of "shaping"). We appreciate the reviewer highlighting this, as we believe it strengthens our submission.
> >
> > > VQ vs SQ.
> >
> > We agree with the reviewer that it is important to compare with commonly used quantization techniques. However, we would like to point out that Linear4bit/Linear8bit without the use of LoRA is essentially regular scalar quantization to a linear range without any Hessian correction, which we compare to in our RTN baseline. It might have not been clear enough that such quantization procedures are still commonly used, and have slightly adapted the text to make this more clear. We did use our own implementation of RTN instead of using the quantizer provided by huggingface, but do not see any reason why either would lead to significantly better or worse results. More importantly, GPTQ and Quarot use Hessian corrections and are considered stronger baselines. In our comparison, we find that PVQ outperforms all of the above in terms of performance over bits per weight and bits per activation.
> >
> > > Optimality of Beta quantizer.
> >
> > To clarify, our method quantizes the amplitude by using the quantile function to map evenly spaced probabilities from the uniform distribution to values on the target distribution (in our case the Beta distribution). By picking the centers of the evenly spaced portions as our quantization points, we ensure that each quantization point in expectation contains equal probability mass and thus the quantization error is minimized. We evenly distribute the quantization error across the intervals, which quantizes the data to _exactly_ those points that minimize the error, given the assumed (Beta) distribution. This indeed requires a non-linear mapping (the CDF/quantile function of the Beta) of the linear range, but this is precisely the (only) nonlinear mapping that minimizes quantization error. We hope the reviewer agrees with the above clarification. If not, could the reviewer be precise in pointing out where he/she things we may be mistaken?
> >
> > > Relation to [2]
> >
> > We thank the reviewer a lot for this reference, which we now include it in the paper. Indeed, this [2] also notes that X/(X+Y) with chi-squared distributed X and Y results in a Beta distributed random variable. We do use a slightly more general notion which considers the sum of not 2 but G chi-squared distributed random variables. Although this is fairly basic statistics, we still deem it highly relevant in this particular context since it prevents having to use and try heuristics (e.g. min/max, symmetric/non-symmetric linear ranges, etc.), which are often occur in similar settings in literature.

---

> > > ### Comment · Reviewer_BTWr · 2024-11-25
> > >
> > > Coherence. If the pre-coherence data is approximately uniformly spherically distributed, what is the purpose of the coherence processing?
> > >
> > > Optimal scalar quantizer for the Beta distribution. "By picking the centers of the evenly spaced portions as our quantization points, we ensure that each quantization point in expectation contains equal probability mass and thus the quantization error is minimized". This is incorrect.

---

> > > > ### Author Response · Authors · 2024-11-25
> > > >
> > > > > Coherence processing
> > > >
> > > > Coherence processing does not modify the distribution of the weights. It’s just a change of coordinate system, which rotating the weights by some orthogonal matrix. Of course this means that the distribution of the _marginals_ changes, which does indeed make it appear more Gaussian when you plot it, and improves the quantization when you assume factorized distribution over each element. We expect coherence processing not to make much of a difference for PVQ on the directions, unless there is some L1-L2 distortion is improved. Yet, in terms of amplitudes we do see the empirical weight distributions to more consistently follow the Beta distribution. This is evidenced by the additional experiments we provide in Appendix D.
> > > >
> > > > > Beta quantizer
> > > >
> > > > Indeed, we agree with the reviewer that our ampltiude quantization is not necessarily optimal in the L2 sense, which would be the Lloyd-Max quantizer. We are sorry for not realising and clearly stating this before yet. It remains that our quantization scheme is expected to be a very effective way of quantizing Beta distributed amplitudes. In particular, because the scheme does not rely on any optimization/search and no additional hyperparameters are introduced. We thank the reviewer for raising the point on optimality, which is critical and expect to have fixed now by removing optimality claims from the paper.

---

> ### Comment · Reviewer_BTWr · 2024-11-25
>
> Coherence processing wouldn't modify the distribution only if the distribution is invariant to rotations. When you use this processing, you make the data seem isotropic and thus you cannot improve quantization due to the structure of the data. Therefore, the only savings you get compared to scalar quantization is the better packing in higher dimensions. You probably want to clarify this point in the paper.
>
> Other than that I have no additional comments.

---

> > ### Comment · Reviewer_BTWr · 2024-11-28
> > **Change in rating**
> >
> > I increased the overall rating to 6. This increase reflects my belief that the authors can rewrite the paper to address the issues raised in the discussion regarding presentation, motivation, and the problem with the declared optimality of the scalar quantizer. Other than this, the methodology seems reasonable and therefore the results are of sufficient interest.

---

### Official Review · Reviewer_8P83 · 2024-11-04

**Soundness:** 3
**Presentation:** 1
**Contribution:** 3
**Rating:** 8
**Confidence:** 3

**Summary:**

The paper addresses the problem of quantization in llms. They introduce PVQ which is a  codebook-free and search-free vector  quantization scheme .

**Strengths:**

-

**Weaknesses:**

-please provide proper  referencing for  “coherence processing” on page 2.
 - on page 3, the   \hat{W} is not well introduced. how is it differen from W ? also \hat{W} should be bold since it is a matrix.
- move the legend in fig 2
- introduce all sets and params before their first appearance: S_D,k , P_D,K not introduced in  section 2.4. sets are better be  distinguishable from matrices  , you could use  caligraphic S and P for  sets.

define “G”  in eq 7.

- scalars better be represented by small letter. W for number of  words can be mistaken with W for weights.

-please address the following  typos/grammatical errors:

“we achieves”, “

“compared to
compared methods”

“maximised”

“, making use for quantization costly.”

“The table can be precomputed a table of N”

“quantised” in fig 6

“On top” , “ to also quantize gains also enabling”

- this is not undestandable:  “to which we refer for a more detailed description of this for attention and fullyconnected layers found in most” - L1 ball should be $l_1$ - E8 ,RTN are not properly intriduced.

-Figure 3 seems to be quantization on a Gaussian source not a neural network. provide mean and variance of the distributon of the Gaussian.

**Questions:**

What is the superiority of your  quantization scheme over the renowned  1.58   llm  quatization scheme?

---

> ### Author Response · Authors · 2024-11-19
>
> Thank you for your feedback and help to improve the paper. We are very glad with the positive review and recommending the paper for acceptance with a rating of 8.
>
> > Coherence processing.
>
> We use the coherence processing originally proposed in Chee et al. (2024), and is used as a pre-processing step in several recent quantization methods, including Quarot (Ashkboos et al., 2024). In coherence processing, weights are rotated through orthogonal matrices (e.g. random Hadamard rotation matrices), which can be undone easily using the transpose. This is efficient and does not alter the output of the network. We have improved our description of coherence processing, fixed the formulas, and properly referenced the relevant prior work.
>
> > E8 and RTN
>
> The E8 method refers to Tseng et al., 2024. RTN refers to naive quantization by rounding weights to the nearest scaled integers. RTN is a standard baseline in quantization and often included in literature as the baseline for naive quantization. We have added a brief description of the methods in the main text to ensure the paper remains self-contained.
>
> > Gaussian source
>
> Most experiments are done on actual LLMs, but in Sec. 3.2 we compare quantization methods to an idealised Gaussian source to empirically measure signal-to-noise ratios. We use the standard multivariate Gaussian for this, zero mean and identity covariance. We thank the reviewers for pointing this out and have added this explicitly in the text.
>
> > Compared to 1.58 bits paper
>
> We focus exclusively on post-training quantization (PTQ). While impressive, BitNet b1.58 is a train-time quantization method and to our knowledge can not successfully be applied post-training. Quantizing the models considered in our work using BitNet b1.58 would require expensive retraining from scratch, which in turn would require access to original training data and vast amounts of computation. Although it would be interesting to also consider PVQ at train time in which case BitNet b1.58 would be a good baseline, which relies on simple scalar quantization, we consider quantization-aware training out of scope for this paper.
>
> Thank you for the typos, we’ve been through the manuscript and fixed them.

---

> > ### Comment · Reviewer_8P83 · 2024-11-21
> >
> > Thank you for adressing my concerns, I have no additional comment.

---

### Author Response · Authors · 2024-11-28

Dear ACs and Reviewers,

We appreciate the valuable feedback provided by the reviewers. Most concerns raised were related to minor issues in the description of the method, and reviewers largely recognized our work as an important contribution.

In particular, we thank Reviewer 8P83 for scoring the paper an 8 and Reviewer BTWr for their extensive review and insightful comments.

However, we do not fully understand why Reviewers 'T3cg' and '1xev' assigned scores of 3 to the paper. The sole reason for their rejection appears to be the lack of comparisons with Quip# [1] and QTIP [2], two vector quantization methods that do not account for activation quantization. For weight-only quantization, we were able to provide a comparison during the rebuttal period (see our comments). Although, for reasons outlined in the same comment, this is not a perfect apples-to-apples comparison, it provides sufficient evidence to show that our method achieves comparable performance at 3-4 bits per weight.

We are making every effort to conduct a more precise comparison for inclusion in the paper. However, this has proven more challenging than initially anticipated due to specific issues with Hugging Face versions and the particular choices of models and calibration datasets used in [1, 2]. Despite this, we believe that even without such a comparison, our paper presents strong evidence of outperforming the most relevant state-of-the-art quantization methods.

Furthermore, while our method surpasses most existing methods for weight-only quantization, achieving the best results in this regime is not the sole focus of our work. Our approach, motivated by recent insights into the spherical geometry of LLM weights, provides a significant step forward toward the quantization of both weights and activations, particularly because of the implicit-codebook and being search-free. We are confident that our work represents an important and meaningful contribution to the machine learning community.

Kind regards,

The Authors


[1] Tseng, Albert, et al. "Quip#: Even better LLM quantization with hadamard incoherence and lattice codebooks." arXiv preprint arXiv:2402.04396 (2024).

[2] Tseng, Albert, et al. "QTIP: Quantization with Trellises and Incoherence Processing." arXiv preprint arXiv:2406.11235 (2024).

---

> ### Comment · Reviewer_T3cg · 2024-11-28
>
> I am very regretful about the authors' response.
>
> I strongly disagree with the authors' following comment: "Despite this, we believe that even without such a comparison, our paper presents strong evidence of outperforming the most relevant state-of-the-art quantization methods."
>
> It is very trivial that vector quantization outperforms scalar uniform quantization, because vector quantization can consider the distribution of weights. **So, claiming the efficacy of the proposed method based on comparisons with scalar uniform quantization is totally unconvincing.**
>
> As I repeatedly mentioned in my response, the proposed PVQ method is the vector quantization method, so comparison should be done with existing vector quantization methods such as QuIP#, QTIP, and others.
>
> If the authors cannot provide fair comparison with existing vector quantization works due to certain reasons (e.g., Hugging Face versions), this means that this paper is not ready for the publication.

---

### Author Response · Authors · 2024-12-03

Dear ACs and reviewers,

We sincerely appreciate the valuable feedback provided by the reviewers. Most concerns raised were related to minor issues in the description of the method, and reviewers largely recognized our work as an important contribution to the field.

In light of this, we were surprised by reviewer 1xev’s decision to lower their score to a 1, whilst other other reviewers provided positive evaluations and appeared largely satisfied with our responses. Notably, reviewer BTWr increased their score from a 5 to a 6, and another reviewer 8P83 maintained their score of 8. Reviewer T3cg rated our work with a 3, primarily due to the lack of comparisons with recent vector quantization methods. To address this concern, we provided a comparison with Quip# and QTIP. While the initial comparison was not entirely apples-to-apples, we can share that we have since repeated the experiment using the same Llama 3 model and the RedPajama dataset and obtained similar results. The updated results remain similar at 3–4 bits, though our method did not outperform QTIP, which achieved a superior average downstream performance (0.70 > 0.68 at 4 bits).

It is worth emphasizing that QTIP is a highly state-of-the-art method; however, it is designed exclusively for weight-only quantization, whereas our method supports the quantization of both weights and activations.

In line with the reviewer guidelines, we hope that not achieving state-of-the-art performance compared to recent vector quantization methods in their specialized domain (i.e., weight-only quantization) will not constitute grounds for rejection. Importantly, these vector quantization methods do not address the quantization of weights and activations together—a key capability of our approach. Our proposed PVQ method encodes vectors of weights and/or activations into integers, enabling matrix multiplication (matmul) operations that operate on integers. This facilitates efficient quantization and reduces the memory required to store model weights and activations during forward-pass computation, all while demonstrably maintaining high performance in terms of perplexity and downstream task performance. This feature is highly relevant from a practical perspective and, to the best of our knowledge, has not been demonstrated before.

We would like to thank the reviewers once again for their constructive feedback.

We think that overall our main contributions are:
- Considering quantization of LLM weights using a spherical code, motivated by recent insights into the spherical distribution of LLM weights.
- Adapting PVQ, a well-known compression algorithm, to the context of LLM quantization and extending it with coherence processing, Hessian-based corrections, and our proposed scale quantization.
- Providing both theoretical and empirical evidence on the distribution of grouped normalized rotated weights.
- Undertaking significant engineering efforts to enable efficient parallel implementation of PVQ with CUDA/C++ hardware acceleration, including improvements in the algorithmic complexity of PVQ subroutines and a custom implementation of arbitrary precision integer arithmetic in CUDA.
- Demonstrately providing a highly performant quantization method applicable to both weights and activations.

and believe that considering this format makes an interesting contribution to the field of quantization.

---

### Meta-Review · Area_Chair_US8y · 2024-12-21

**Metareview:**

**Summary:**
This paper proposes application of pyramid vector quantization (PVQ) to compression of weights and activations in large language models (LLMs).

**Strengths:**
This paper discusses an important and timely subject of compressing LLMs, and the proposal based on PVQ is interesting.

**Weaknesses:**
- Most reviewers criticized lack of comparison of the proposal with existing vector quantization methods.
- Most reviewers evaluated quality of the presentation of this paper, in its current form, negatively.

**Reasons:**
Besides the lack of comparison with existing vector quantization methods, I am concerned the most with the quality of the presentation, which would require much more careful proofreading to have the manuscript to get accepted. Below is a relatively long list of points which I think would require revision.

Additional points:
- Page 1, line 22: we achieve(s); (p → P)areto-optimal
- Page 1, lines 23-4: compared to (compared) methods
- Page 1, line 38: has been (been)
- Page 2, lines 4, 45-6; page 3, line 10: pyramid vector quantization → PVQ
- Page 2, line 8: in (a) some chosen
- Page 2, lines 7-9: While the authors argue quantization of weights $W$ and activations $x$, they only introduce $\hat{W}$ and not $\hat{x}$.
- Page 2, line 9: A common conve(rs → nt)ion
- Page 2, line 12: The expectation $\mathbb{E}_{x}[xx^T]$ should never be equal to the empirical mean. One would also need the prefactor $1/N$.
- Page 2, line 14: the (layer-wise output → output error?) at each layer
- Page 2, line 17: "Equation" can be removed.
- Page 2, line 26: higher (number of) dimensions
- Page 2, line 36: "shape" might be better termed as "direction".
- Page 2, lines 38-9: (Kosson et al., 2023) → Kosson et al. (2023)
- Page 2, line 43: that reparameterize(s)
- Page 3, line 2: Chee et al. (2024) → (Chee et al., 2024)
- Page 3, lines 3, 5: The formulas $W=UWV$ and $\hat{W}=V^T\hat{W}U^T$ do not make sense.
- Page 4, line 6: To quantiz(ing → e)
- Page 4, line 14: sum to (1 → $K$)
- Page 4, line 17: $c\in[1,\ldots,N(D,K)-1]$ → $c\in[1,N(D,K)]$
- Page 4, line 25: that allow(s)
- Page 4, line 27: $C_{2,7}=[0,27]$ → $C_{2,7}=[1,28]$; Figure 2 should be corrected accordingly.
- Page 4, lines 27-30: I did not understand the sentence "The table can be precomputed a table ...". It might have been meant "The table of $N(d,k)$ for ... can be precomputed and reused ...".
- Page 5, line 4: (in) Section 3.2.
- Page 6, lines 20-1: "until convergence" should be moved just after "$|w_i|-K$"; One should refer not to Section 2.4 but to Section 2.5.
- Page 6, line 26: The formula for $s$ should be $s=\hat{w}^Tw/\|\hat{w}\|^2$.
- Page 6, line 30; Section 3.4: As Reviewer BTWr questioned, as opposed the proposal by the authors here, it is known that the quantization that is equispaced in the CDF values is not optimal in terms of the mean squared quantization error. See, e.g., Zador, IEEE Trans. Info. Theory, volume 28, pages 139-149, 1982.
- Page 6, equation (8): Please think about most elegant notation here, as is written.
- Page 6, lines 46-7: custom (arithmetic → arbitrary) precision arithmetic
- Page 7, line 7: coefficients → parameters
- Page 7, Theorem 3.1: Why do not the authors write $s_h=\|v_g\|^2/\|w\|^2$? The second parameter of the beta distribution should be $\frac{D(G-1)}{2}$, not $\frac{G(D-1)}{2}$. The statement of this theorem is well known: See, e.g., Ross, *A First Course in Probability*, 10th ed., Pearson, 2020, where Example 3b in Section 6.3.2 states that a sum of iid zero-mean Gaussians is chi-squared and thus gamma, and where Example 7c in Section 6.7 states that $X/(X+Y)$ for independent gammas $X,Y$ is beta.
- Page 7, Figure 5 caption: closely match(es)

**Additional Comments On Reviewer Discussion:**

The review scores exhibited a relatively large split, from 8 to 1. After the author rebuttal we had relatively long discussion between the authors and the reviewers, which would suggest that this paper is not yet of good enough quality to support acceptance.

---

### Decision · Program_Chairs · 2025-01-22

Reject